# Uncovering the Spectral Bias in Diagonal State Space Models

**Ruben Solozabal**
MBZUAI
ruben.solozabal@mbzuai.ac.ae

**Velibor Bojkovic**
MBZUAI
velibor.bojkovic@mbzuai.ac.ae

**Hilal AlQuabeh**
MBZUAI, RIKEN AIP
hilal.alquabeh@mbzuai.ac.ae

**Kentaro Inui**
MBZUAI, RIKEN AIP
kentaro.inui@mbzuai.ac.ae

**Martin Takáč**
MBZUAI
Takac.MT@gmail.com

## Abstract

Current methods for initializing state space models (SSMs) parameters mainly rely on the *HiPPO framework*, which is based on an online approximation of orthogonal polynomials. Recently, diagonal alternatives have shown to reach a similar level of performance while being significantly more efficient due to the simplification in the kernel computation. However, the *HiPPO framework* does not explicitly study the role of its diagonal variants. In this paper, we take a further step to investigate the role of diagonal SSM initialization schemes from the frequency perspective. Our work seeks to systematically understand how to parameterize these models and uncover the learning biases inherent in such diagonal state-space models. Based on our observations, we propose a diagonal initialization on the discrete Fourier domain *S4D-DFouT*. The insights in the role of pole placing in the initialization enable us to further scale them and achieve state-of-the-art results on the Long Range Arena benchmark, allowing us to train from scratch on very large datasets as PathX-256.

## 1 Introduction

State space models (SSMs) have recently emerged as a principled and scalable means of modeling long sequences across diverse domains, including image processing [1, 2, 3], time-series forecasting [4], and natural language understanding [5, 6, 7]. At their core, SSMs perform a sequence-to-sequence mapping via a long-range convolution kernel that is parameterized under a continuous-time linear dynamical system. This formulation captures dependencies at multiple timescales and admits strong stability guarantees. Crucially, the "HiPPO matrix" [8] provides a principled method to structure state transition matrices that performs online compression of the input stream by projecting onto an orthogonal polynomial basis. However, naively propagating this dense state through a sequence of

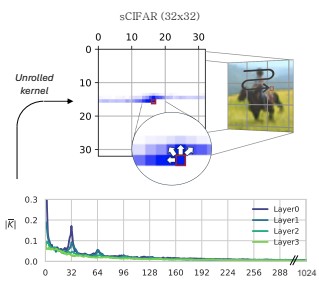

Figure 1: The kernels learned by S4D on `sCIFAR` present a "local attention" profile. When unrolled, these kernels align with positions corresponding to the vicinities of the pixel being attended.

39th Conference on Neural Information Processing Systems (NeurIPS 2025).

length $L$ incurs in $\mathcal{O}(N^2 L)$ cost, where $N$ is the state dimension, making direct implementation impractical for very long sequences. To overcome this bottleneck, the Structured State Space Sequence (S4) [9] model introduces a normal plus low-rank decomposition of the transition matrix, which reduces the computational complexity while maintaining the expressive capacity [10].

Building on the foundations of S4, recent diagonal variants [11, 12, 13, 14], have shown that restricting the state matrix to be diagonal still preserves the performance. By introducing complex-valued initialization schemes [12], these diagonal SSMs achieve performance comparable to the original S4 architecture while being more computationally efficient, as the computation of the kernel simplifies to a Vandermonde matrix multiplication. Nevertheless, while the S4 framework has a mathematical interpretation for addressing long-range dependencies, the efficacy of its diagonal variants remains theoretically unexplained.

In this work, we take a step further by investigating the initialization of diagonal SSMs from a frequency perspective [15]. We aim to uncover how the initialization of the state matrix, particularly the distribution of the imaginary part of its eigenvalues, directly influences the system's ability to capture long-range dependencies. Our analysis reveals that current schemes highly depend on the input sequence exhibiting a well-defined inherent timescale that aligns with the model's discretization step $\Delta$ [10]. Since such alignment is unknown a priori, existing models often compensate by spreading the poles across a wide frequency range, resulting in over-parameterization [16], non-uniform spectral sensitivity, and aliasing artifacts.

Building on this analysis, we introduce S4D-DFouT, an initialization in the discrete domain that explicitly ensures a uniform coverage of the frequency spectrum independently of the selection of the discretization step $\Delta$. By placing poles directly in the discrete domain, we can reduce the sensitivity these models present to the frequency response without compromising its ability to capture both short- and long-range effects.

To summarize, the key contributions of this paper are threefold:

- **Frequency-aware analysis of Diagonal SSMs:** The theoretical understanding of the roles played by the initialization schemes on diagonal SSMs is still lacking. Therefore, we systematically describe diagonal SSMs from a frequency perspective. We analyze how different initialization schemes affect the model's ability to represent temporal dependencies (Section 3).
- **Proposal of S4D-DFouT initialization:** Our observations indicate that current models are highly sensitive to the discretization. We propose a novel approach, S4D-DFouT, directly in the discrete domain that explicitly designs the pole distribution to optimize spectral coverage. This strategy eliminates the existing coupling between the decay and frequency selection, resulting in a more robust and easy-to-scale initialization. (Section 4).
- **SSMs initialization synchronization enhances model efficiency:** Finally, we propose a synchronized initialization across all the SSMs in a given layer. By coordinating their pole placements, the entire layer can be jointly initialize. Our experiments demonstrate that such frequency-aware synchronization enables diagonal SSMs to scale more effectively and outperform previous methods on Long Range Arena (LRA) [17] enabling to train from scratch on `PathX-256` (Section 5).

## 2   Related literature

Initial approaches to sequence modeling, such as recurrent neural networks (RNNs) and long short-term memory (LSTM) [18] models, have achieved notable success but encountered significant limitations when modeling long-range dependencies, primarily due to vanishing and exploding gradients [19, 20]. These challenges motivated the search for efficient alternative architectures, such as Legendre Memory Units (LMUs) [21], which introduce a non-trainable continuous-time SSM parameterized with Legendre polynomial bases to encode long-term memory. Building on the LMU, the HiPPO framework provided a principled method to construct state matrices with trainable state matrices, leading to a successful family of long-range convolution models [9, 14, 22, 23].

Diagonal variants of SSMs have recently emerged as a simplification of the DPLR-based S4 architecture [11, 12, 13]. The Diagonal State Spaces (DSS) [11] method empirically demonstrated that simply removing the low-rank term from the HiPPO DPLR representation to form a diagonal HiPPO matrix preserves performance. Building on this insight, the S4D framework [12] systematically explores diagonal initialization. Nevertheless, while the S4 framework has a mathematical interpretation

for addressing long-range dependencies, the efficacy of its diagonal variants remains theoretically unexplained.

Most recently, [24] provided a data-centric perspective on SSM initialization. Their work emphasizes the importance of aligning model initialization with the autocorrelation structure of the input data. We push beyond this data-driven criterion by investigating the initialization of diagonal SSMs from a frequency perspective. We observe that current initialization schemes (whether based on HiPPO inverse-frequency laws, or linear grids) drive the model to learn convolutional kernels whose resonance aligns with the principal frequential components of the data. As illustrated in Fig. 1, the kernels learned by SSMs act as localized, image-sized filters, effectively aggregating spatially local features rather than capturing truly global, long-range interactions. Previous works [25] have leveraged exactly this effect on Transformers [26, 27] to perform well on LRA. By combining attention with an exponential moving average, those works inject a built-in position-aware encoding mechanism that facilitates learning such tasks.

Our findings show that current initialization schemes rely heavily on coincidental alignment between the dominant timescales in the input and the poles in the system. Since such alignment is rarely known a priori, existing models often compensate by spreading poles across a wide frequency range, resulting in over-parameterization. Recent works have shown that large portion of the modes in these models can be pruned with a minimal effect in the model performance [16].

Building on these frequency-domain insights, we developed the S4D-DFouT initialization. A universal initialization directly in the discrete domain that decouples the interdependency between the decay rate and the oscillation frequency. This design not only liberates from the sensitivity in the selection of $\Delta$, but also produces kernels that maintain consistent expressivity across a wide range of frequencies. Leveraging DFouT, we demonstrate, for the first time, successful training from scratch on `PathX-256`, a task that previous state-of-the-art methods could only tackle after extensive self-pretraining [28].

## 3  Preliminaries

In this section, we briefly introduce the diagonal SSM and the problem setting we considered throughout this paper. Specifically, we analyze the single-input single-output (SISO) variant of the S4 model, where the state transition matrix is defined over the field of complex numbers $\mathbb{C}$, while the model outputs reside in the real domain $\mathbb{R}$. The system dynamics is described with the following equations:

$$\frac{d}{dt} h(t) = \Lambda\, h(t) + B\, x(t), \qquad y(t) = \Re\big(C^\top h(t)\big), \quad t > 0. \tag{1}$$

Here, $t \in \mathbb{R}_{\geq 0}$ is the continuous time variable, and $\Re(\cdot)$ represents the real part of the complex entry vector. The input and output signals $x(t), y(t) \in \mathbb{R}$ are real valued, while the hidden state $h(t) \in \mathbb{C}^N$ evolves in the complex domain. In particular, we consider the setting where the state matrix is diagonal $\Lambda = \mathrm{diag}(\lambda_1, \lambda_2, \ldots, \lambda_N)$. Under these settings, the input-output relation in (1) is explicitly given by the integral

$$y(t) \;=\; \int_0^t \Re\big(C^\top e^{\Lambda\, s} B\big)\, x\big(t - s\big)\, \mathrm{d}s, \tag{2}$$

i.e. the continuous-time SSM can be represented as a continuous convolution $y(t) = (x * K)(t)$ of the input signal with the kernel function $K = \Re\big(C^\top e^{\Lambda\, s} B\big)$.

### 3.1  Discretization

To obtain a discrete-time version of the continuous diagonal SSM introduced above, one can apply standard discretization techniques. Common approaches include the forward and backward Euler schemes, bilinear transform, and zeroth-order hold (ZOH) method. In this work, we adopt the latter, which yields the following discretized parameters from the continuous-time system:

$$\overline{\Lambda} \;=\; \exp(\Delta\Lambda), \quad \overline{B} \;=\; (\Delta\Lambda)^{-1}\big(\exp(\Delta\Lambda) - I\big)\,\Delta B. \tag{3}$$

Under this discretization, the state update and output equations take on recursive form and become:

$$h[l] = \overline{\Lambda}\, h[l - 1] + \overline{B}\, x[l], \quad y[l] = \Re\big(C^\top h[l]\big), \quad l = 0, \ldots, L - 1, \tag{4}$$

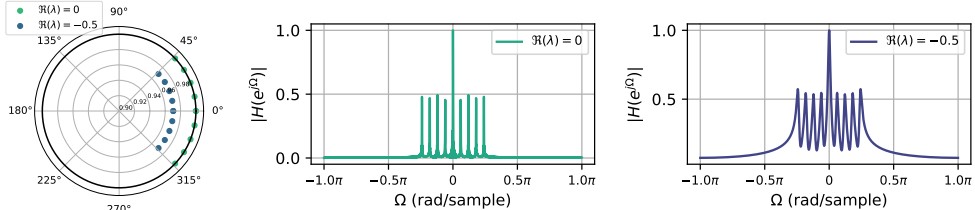

Figure 2: Frequency response of a diagonal SSM. **Left:** An example of pole configuration (i.e. state matrix entries $\lambda_1, \ldots, \lambda_N$) for two discrete SSM systems of order $N = 10$ exhibiting conjugate symmetry. Following equation (8) we plot the corresponding frequency responses in dependence of the angular frequency of the corresponding discretized systems with $\Delta = 0.1$ (**center** and **right** figures). The system with $\Re(\lambda) = 0$ (**center**) presents a narrower response around the resonant frequencies, whereas in the system with $\Re(\lambda) = -0.5$ (**right**) those are wider.

where the input $x = (x[0], \ldots, x[L - 1])$ now becomes a sequence of numbers of length $L$. The corresponding convolutional or kernel-based view of the model is:

$$y[l] = \Re\big((x * \overline{K})[l]\big), \quad \text{with} \quad \overline{K} = \big(C^\top \overline{B}, C^\top \overline{\Lambda B}, \ldots, C^\top \overline{\Lambda}^{L-1} \overline{B}\big). \tag{5}$$

### 3.2 Frequency interpretation of the kernel

To understand the frequency characteristics of the discrete convolution kernel $\overline{K}$, it is useful to decompose it into a linear combination of basis kernels, where each basis kernel corresponds to a single mode (entry) of the state-space model and captures a distinct frequency component of the system. We define the basis kernels $\overline{K}_n[l]$ as

$$\overline{K}_n[l] := \overline{\lambda}_n^l \overline{B}_n, \quad l = 0, \ldots, L - 1, \tag{6}$$

where $\overline{\lambda}_n = \exp(-\Delta\,\alpha_n + i\,\Delta\,\omega_n)$ is the $n$th eigenvalue (entry) of the discretized state matrix $\overline{\Lambda}$, and $\overline{B}_n$ is the corresponding component of the input projection vector $\overline{B}$. The full kernel $\overline{K}[l]$ is then obtained as a linear combination of these basis elements, weighted by the components $C_n$ of the output projection vector $C$:

$$\overline{K}[l] = \sum_{n=0}^{N-1} C_n \overline{K}_n[l] = \sum_{n=0}^{N-1} C_n \overline{B}_n \overline{\lambda}_n^l. \tag{7}$$

From the expressions above, it follows that each individual basis kernel $\overline{K}_n$ corresponds to a scaled damped complex exponential with decay rate $\Delta\,\alpha_n$ and discrete angular frequency $\Omega_n := \Delta\,\omega_n$. As such, each channel $n$ can be seen as a single-pole filter that resonates at frequency $\Omega_n$ with its temporal decay determined by $\Delta\,\alpha_n$.

The discrete-time frequency response of the kernel $\overline{K}[l]$ is computed via the discrete-time Fourier transform as

$$
\begin{aligned}
H(e^{i\,\theta}) &:= \sum_{l \geq 0} \overline{K}[l] e^{-i\,\theta\,l} = \sum_{n=0}^{N-1} C_n \overline{B}_n \sum_{l \geq 0} e^{(-\Delta\,\alpha_n + i\,(\Omega_n - \theta))\,l} \\
&= \sum_{n=0}^{N-1} \frac{C_n \overline{B}_n}{1 - e^{-\Delta\,\alpha_n}\,e^{i\,(\Omega_n - \theta)}}.
\end{aligned}
\tag{8}
$$

The last expression shows that each component contributes a resonant peak centered on $\theta = \Omega_n$, with the amplitude controlled by $C_n \overline{B}_n$ and the bandwidth governed by the decay rate $\Delta\,\alpha_n$. When $\Delta\,\alpha_n$ is small, the resonance is sharp, indicating a narrow-band frequency response, while when the decay rate is large, the frequency response becomes broader (Fig. 2).

### 3.3 Parameterization and Initialization of Diagonal State Matrices

The trainable parameters of the diagonal state-space models are the eigenvalues $\Lambda \in \mathbb{C}^{N \times N}$, and the input/output projections $B, C \in \mathbb{C}^N$, with hidden state initialized as $h(0) = \mathbf{0}$. In practice, $B$

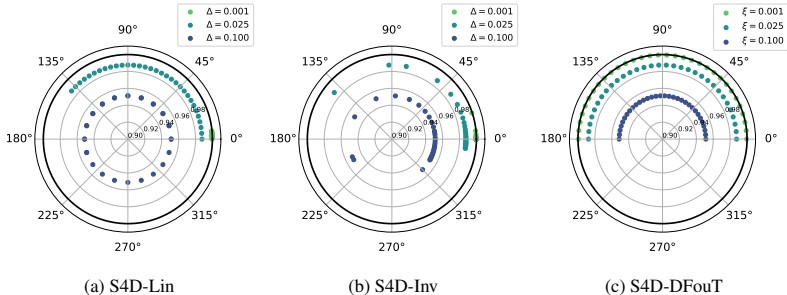

Figure 3: Visualization of the poles configuration obtained on S4D initializations for a system of order $N = 32$ upon the selection of different discretization or decay steps $\Delta, \xi \in \{0.001, 0.025, 0.1\}$. In the S4D-Lin and S4D-Inv schemes, $\Delta$ controls both the decay (*radius*) and resonant frequencies (*angle*) of the system. By contrast, S4D-DFouT is designed to ensure even spectral coverage at any decay rate.

is often initialized as a vector of ones, while the eigenvalues $\Lambda$ are initialized according to different proposed schemes [12].

In the original S4 model, initialization is done in the continuous-time domain where a structured matrix $\Lambda$ is selected (e.g. from HiPPO-LegS), then discretized with a chosen parameter $\Delta$, to obtain $\overline{\Lambda}$. Furthermore, several S4D variants provide simplified diagonal initializations [11, 12]:

**S4D-LegS.** Uses only the diagonal part $\Lambda^{(D)}$ of the HiPPO-LegS matrix $A$ when the later is put into diagonal plus low-rank form $A = \Lambda^{(D)} + P\,P^\top$, dropping the low-rank correction.

**S4D-Inv.** Eigenvalues of the state matrix $\Lambda$ are initialized following an inverse-frequency law for the imaginary parts, with fixed real parts as in the left side of (9).

**S4D-Lin.** Eigenvalues of the state matrix $\Lambda$ are coming from a linearly spaced frequency grid, with fixed real parts, as in the right side of (9).

$$\textbf{S4D-Inv: } \lambda_n = -\frac{1}{2} + i\frac{N}{\pi}\left(\frac{N}{2n+1} - 1\right) \qquad \textbf{S4D-Lin: } \lambda_n = -\frac{1}{2} + i\pi n \qquad (9)$$

For the latter two cases, we make a crucial observation: The diagonal state matrix is initialized in the continuous domain, which during the training is discretized with a learnable parameter $\Delta$. In particular, *the poles of the discretized diagonal system will highly depend on $\Delta$*. This observation serves as a starting motivation for our proposed method.

## 4 Main Results

### 4.1 Motivation

**Entanglement of decay and frequency via discretization.** Existing initialization schemes, such as the ones described in the previous section, are typically defined in continuous time and then discretized before training. This process introduces fundamental coupling between the decay rate and the oscillation frequency through the discretization parameter $\Delta$. Specifically, both the real and imaginary part in the eigenvalues of the discretized system (3) scale linearly with $\Delta$. Consequently, adjusting the temporal resolution $\Delta$ alters both the decay and resonant frequencies of the system, making the model's behavior sensitive to the discretization grid. This coupling potentially complicates both model tuning and interoperability, as changing the discretization has unintended effects on the frequency response.

**Limited control over Spectral Coverage.** The frequency components induced by previous initialization strategies often lack direct interpretability or fine-grained control. For instance, S4D-Inv spreads frequencies according to an inverse-law, clustering models disproportionately at low frequencies, while S4D-Lin distributes them linearly but with a fixed decay rate across all modes (Fig. 3 (a), (b)). These choices may not align well with the actual spectral content of input signals, especially when dealing with broadband inputs or tasks requiring uniform resolution across frequency bands. Furthermore, the decay rates in these schemes are typically hardcoded or indirectly determined via

the continuous-time representation, offering little flexibility in shaping the temporal memory profile of the model without extensive hyperparameter tuning.

In summary, the discretization step *entangles decay and frequency components in a way that complicates interpretability and control, while also making the initialization sensitive to the choice of time step $\Delta$*. Moreover, these schemes often *impose rigid spectral structures that may not align with the needs for real-world tasks*. These limitations motivate the need for an initialization approach defined directly in the discrete domain.

## 4.2 *S4D-DFouT*: A Universal Initialization Scheme of Discrete SSMs

As we argued in the previous section, a central challenge in diagonal SSMs is choosing the discretization step $\Delta$ when the data's intrinsic timescale is unknown. Common practices sample $\Delta$ at random from a log-uniform range $\Delta_h \sim \text{Log}\mathcal{U}(\Delta_{\min}, \Delta_{\max})$, to hedge over possible time resolutions. While this increases the odds that at least some modes align with the true dynamics, it also inflates the model's parameter count and yields uneven coverage of the frequency axis. Furthermore, due to the initialization in the continuous domain, the choice of $\Delta$ becomes critical: if $\Delta$ is too small, all modes collapse into a narrow low-frequency band; if it is too large, modes fold onto one another and alias, thus reducing spectral diversity as the following Lemma suggests.

**Lemma 1.** *After discretizing with step $\Delta$, a continuous pole $\lambda = -\alpha + i\omega$ $(\alpha > 0)$ becomes $\overline{\lambda} = \exp(-\alpha\Delta + i\Omega)$ with $\Omega \equiv \Delta\omega \pmod{2\pi}$. Non-aliasing condition states that distinct analogue frequencies $\omega_m \neq \omega_n$ map to distinct digital frequencies $\Omega_m \neq \Omega_n$. In particular, resonant frequencies would not alias with any other if $|\Delta\omega| < \pi$, the Nyquist bound. Thus, choosing the imaginary parts so that $|\Delta\,\Im(\lambda_n)| < \pi$ for all $n$ guarantees an alias-free frequency coverage.*

To address these problems, we propose a new initialization scheme directly in the discrete domain.

**S4D-DFouT: Single SSM initialization.** We propose a novel initialization scheme, S4D-DFouT, which directly constructs the discrete-time state matrix with diagonal entries

$$\textbf{S4D-DFouT: } \overline{\lambda}_n = \exp\left(-\frac{\xi_n}{2} + i\frac{2\pi n}{N}\right), \quad n = 0, 1, \ldots, N - 1, \tag{10}$$

where $\xi > 0$ is a learnable damping factor. This initialization places all poles uniformly around the unit circle in the complex plane, modulated by a shared exponential decay. The imaginary components $i\frac{2\pi n}{N}$ ensure complete and uniform coverage of the frequency spectrum, while the damping term $-\frac{\xi}{2}$ governs memory retention by controlling how quickly each mode decays over time.

When $\xi = 0$, the resulting state transition matrix becomes unitary, and the associated SSM implements a perfect frequency basis, reducing exactly to the Discrete Fourier Transform (DFT). In this limiting case, the model can exactly represent any circular convolutional kernel of length $N$. Moreover, due to its complete and non-redundant spectral basis, the S4D-DFouT initialization enables the system to act as a universal approximator.

**S4D-DFouT: Layer-wise initialization.** An S4 layer consists of multiple single-input, single-output SSMs operating in parallel, one per feature dimension in the embedding space. Suppose the embedding dimension is $H$, then an S4 layer comprises $H$ diagonal SSMs, each with an internal hidden state of size $N$. To avoid redundant poles and to increase the coverage of angular frequencies of SSMs working together, we synchronize their DFouT initializations by assigning a fixed phase offset to every machine $\{\phi_h\}_{h=1}^H = \frac{2\pi(h-1)}{NH} \subset [0, 2\pi/N)$. Thus, the poles of the $h$-th SSM are

$$\overline{\lambda}_{h,n} = \exp\left(-\frac{\xi_n}{2} + i\left(\frac{2\pi n}{N} + \phi_h\right)\right) = \exp\left(-\frac{\xi_n}{2} + i\frac{2\pi\left(nH + (h-1)\right)}{NH}\right). \tag{11}$$

Across the layers, the collection $\{\overline{\lambda}_{h,n}\}$ forms a uniform grid of size $NH$ on $[0, 2\pi)$, ensuring every resonance band is represented exactly once and preventing multiple machines from competing at nearly identical frequencies.

**S4D-DFouT: Real vs. Complex Inputs.** Ensuring real-valued outputs requires that any complex eigenvalues occur in conjugate pairs. Rather than enforcing this conjugate symmetry directly—an operation complicated in S4's learnable parameterization—the original S4 formulation simply takes

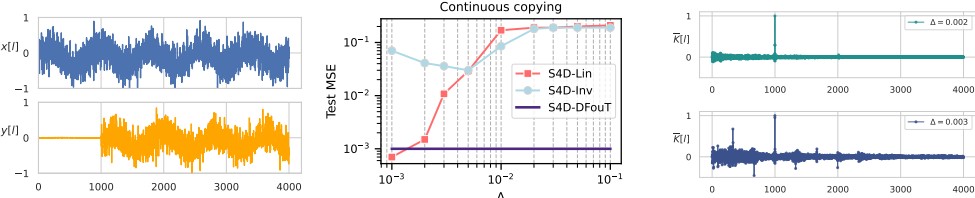

Figure 4: Continuous copying or `Delay` task. **Left:** An example input $x[l]$ and corresponding delayed output $y[l]$. **Center:** Reconstruction MSE for different initializations as a function of $\Delta$. **Right:** The S4D-Lin kernel learned at the theoretically optimal $\Delta = 0.002$ (of similar performance of S4D-DFouT), and its distortion when $\Delta = 0.003$.

the real part of the resulting kernel. In our proposal, we adopt a similar strategy. Furthermore, when the input sequence $x[l]$ is real-valued, hence its $N$-point DFT $X[N]$ exhibits *Hermitian symmetry* $X[N - n] = \overline{X[n]}$, for $n = 1, 2, \ldots, \lfloor \frac{N}{2} \rfloor$, all negative-frequency bins are redundant and the spectrum is fully determined by the *positive* frequencies $n = 0, 1, \ldots, \lceil N/2 \rceil$. Exploiting this fact, one can initialize the diagonal SSM with only $N^+ = \lceil \frac{N}{2} \rceil + 1$ poles distributed on the upper disc $\{ e^{i\Omega} \mid \Omega \in [0, \pi] \}$ instead of the full $2\pi$ range. This "half-plane" S4D-DFouT variant therefore reduces the state dimension and computational cost by a factor of two, yet retains the complete information content of the sliding DFT for real-valued signals (Fig. 3 (c)).

## 5   Experiments

Our experimental evaluation proceeds as follows. First, we introduce a motivating example in the Continuous Copying task. Then, we utilize `sCIFAR` to probe the inductive biases that SSMs exhibit when learning on serialized image data. Finally, we demonstrate the benefits of our S4D-DFouT initialization across the Long Range Arena benchmark [17], and further ablation datasets as the Speech Commands dataset [29]. Details of the experimental settings are provided in the Appendix C.

### 5.1   Continuous copying task

To motivate our frequency-based analysis, we first consider the Continuous copying or `Delay` task, a test of long-term memory where the correct pole placement is critical for success. Here, the model must learn to perform a sequence-to-sequence mapping $\mathbb{R}^{4000} \to \mathbb{R}^{4000}$ where the output is lagged by $\tau = 1000$ steps (Fig 4 Left). The architecture comprises a single SSM of order $N = 1024$ followed by a linear output layer, and is trained on white-noise inputs bandlimited to 1000 Hz.

Successfully solving this task requires the SSM to approximate the ideal delay kernel, which is zero everywhere except at lag $\tau$. In our experiment, we fix the $\Re(\lambda) = 0$ to ensure no decay and center the experiment on the frequency selection. In this setup, we demonstrate that for previous initializations under an inappropriate selection of $\Delta$ fail to capture the desired dynamics (reconstruction MSE error is depicted in Fig 4 Center. In the particular case of S4D-Lin, the solution to this task is almost trivial when $\Delta = \frac{2}{\tau}$, as stated in the following Proposition.

**Proposition 1.** *Let the S4D-Lin convolutional kernel under ZOH discretization be* $\overline{K}[l] = 2\Re(\sum_{n=0}^{N-1} C_n \overline{\lambda}_n^l)$. *If we choose* $\Delta = \frac{2}{\tau}$, *where* $\tau \in \mathbb{Z}_{>0}$, *and take* $C_i = 1$ *for all* $i$, *then* $\overline{K}$ *presents a "spike" at* $l = \tau$, *with* $|\overline{K}[l]| < |\overline{K}[\tau]|$ $(\forall l \neq \tau)$. *Moreover, if we take $C$ to be a trainable parameter in general, the Vandermonde matrix* $(V_{ln}) = (\overline{\lambda}_n^l)$ *is well-conditioned and gradient descent for $C$ converges at a linear rate.*

Moreover, for S4D-Lin any choice of $\Delta > \frac{2}{\tau}$ makes the fundamental period to fall below $\tau$, and no isolated spike at $l = \tau$ can be formed in the kernel. Empirically, this misalignment leads to a large reconstruction error as shown in Fig 4 Right. In contrast, S4D-DFouT successfully manages to reconstruct the ideal delay kernel regardless of the initialization.

Table 1: Accuracy on the Long Range Arena (LRA) benchmark. "✗" indicates computationally infeasible runs, "□" denotes unreported results, and "†" experimentally reproduced in this work. The highest-performing diagonal variant is shown in **bold**. Hyperparameter settings are detailed in Appendix C.8.

| Method | ListOps (2048) | Text (4096) | Retrieval (4000) | Image (1024) | Pathfinder (1024) | PathX-128 (16,384) | PathX-256 (65,536) |
|---|---|---|---|---|---|---|---|
| Transformer | 36.37 | 64.27 | 57.46 | 42.44 | 71.40 | ✗ | □ |
| S4-LegS | 59.60 | 86.82 | 90.90 | 88.65 | 94.20 | 96.35 | □ |
| S4-FouT | 57.88 | 86.34 | 89.66 | 89.07 | 94.46 | ✗ | □ |
| S5 | 62.15 | 89.31 | 91.40 | 88.00 | 95.33 | 98.58 | □ |
| S4D-Lin | 60.52 | 86.97 | **90.96** | 87.93 | 93.96 | ✗ (**96.02**†) | □ |
| S4D-Inv | 60.18 | 87.34 | 91.09 | 87.83 | 93.78 | 92.80 | □ |
| S4D-LegS | 60.47 | 86.18 | 89.46 | 88.19 | 93.06 | 91.95 | □ |
| S4D-DFouT | **61.89** | **87.41** | 90.95 | **88.48** | **94.30** | 94.17 | **87.89** |

## 5.2 Pixel-level image classification

As introduced in Fig. 1, the kernels learned using S4D initialization on `sCIFAR` present a "local attention profile", exhibiting clear peaks at offsets of 32 and being almost zero beyond the first entries. When unrolling these kernels to the original $32 \times 32$ image grid, the peaks align with positions corresponding to the vicinities of the pixel being attended at a certain time step (i.e., local nearby pixels in the adjacent rows). Therefore, we conclude that the model's receptive field learned is a compact band of locally nearby pixels in the image.

We compare this behavior to CNNs, where early convolutional layers specialize in detecting localized features, while deeper layers gradually integrate these local patterns into a more abstract global representation.

To quantify the impact of this effect, we retrain the baseline model progressively shortening the length of the kernel (Fig. 5 Left). As depicted, the model preserves the original accuracy when the kernel is reduced to only 32 coefficients, confirming that the model's performance relies almost exclusively on very local context. Beyond that, the degradation is gradual, with nontrivial accuracy preserved down to surprisingly small kernel sizes.

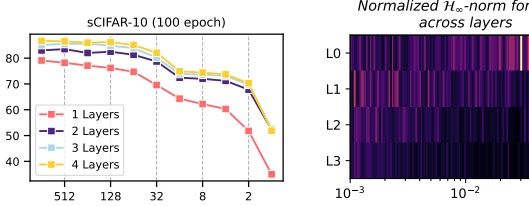

Figure 5: Pixel-level image classification on `sCIFAR`. **Left:** Ablation of the accuracy upon a reduced kernel length and number of trainable S4D-Lin layers. **Right:** Normalized $\mathcal{H}_\infty$-norm for each individual SSM initialized under S4D-Lin, sorted by its operating $\Delta$.

Furthermore, we measure how much each SSM actually contributes to the output by computing its $\mathcal{H}_\infty$-norm [16]. The model initialized with S4D-Lin comprises four layers of 128-dimensional embeddings, yielding a total of 512 SSMs, each initialized with a random discretization step sampled from the range $\Delta \in [10^{-3}, 10^{-1}]$. Strikingly, only those SSMs whose discretization step falls within a narrow subinterval in the first layer exhibit non-negligible $\mathcal{H}_\infty$-norms; the vast majority of them, particularly in deeper layers, remain effectively inactive (Fig. 5 Right). We confirm this in the following experiment. We further ablate the layer's importance by retraining only with the first $k$-layer kept active and replacing all subsequent SSM layers with identity mappings (Fig. 5 Left). In this setup, we observe that almost all of the learned sequential information resides in that very first SSM layer, capturing the critical local pixel-row dependencies, while deeper SSM layers only provide incremental refinement.

## 5.3 Long range arena

The LRA benchmark has been widely adopted to assess a model's capacity for capturing long-range context across diverse sequence lengths and modalities. In this sense, the Fourier-initialized variants have repeatedly underperformed on its hardest tasks, failing to solve challenges like `PathX-128` that other initializations could handle. We show that this shortfall is not an intrinsic limitation of Fourier

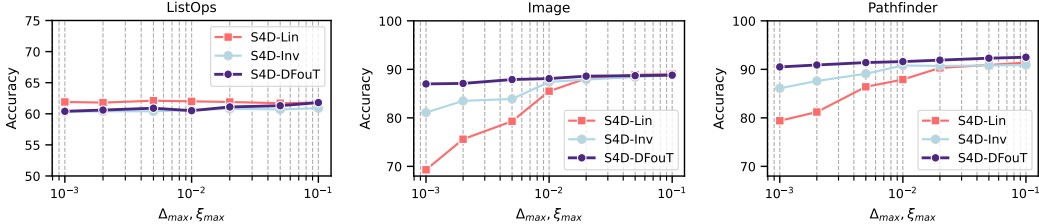

Figure 6: Ablation experiment in LRA. Accuracy upon different initialization schemes for fixed $\Delta_{min}, \xi_{min} = 10^{-3}$ and variable $\Delta_{max}, \xi_{max}$. Datasets `Image` and `Pathfinder` presenting a strong frequential component (see Fig. 7) are highly sensitive to an initialization that captures that frequency.

initialization, but rather of the difficulty of selecting the $\Delta$ hyperparameter. The intuition gained on the "local-attention" profile these kernels exhibit on serialized images provided us a prior on the selection of $\Delta$ to successfully learn this task (marked with "†" in Table 1). These results show that Fourier-based state-space models can equally perform well, provided their discretization aligns with each task's intrinsic timescale.

Furthermore, our proposed S4D-DFouT initialization liberates from these difficulties, and we could directly apply it successfully to scale to harder tasks such as `PathX-256` without any problem-specific tuning. So far, this is the first work to succeed in learning on `PathX-256` from scratch, as previous approaches relied on *self-pretraining* [28] to handle sequences of this length.

### 5.4 Raw speech classification

We also evaluate the method on the Speech Commands, a 35-way spoken-word classification task using raw audio waveforms. Each input is a 1-second signal sampled at 16 kHz. While previous studies typically extract spectral features such as MFCCs [30], state-space models like S4 have demonstrated the ability to learn directly from time-domain inputs. As shown in Table 2, we quantify the performance gains under a zero-shot resampling scenario, where each test waveform is uniformly subsampled to 8 kHz without retraining. In the latter, we surpass the previous initializations by 2 points.

### 5.5 Ablations

In Fig. 6 we perform an ablation study on the effect of upper bounds $\Delta_{max}$ and $\xi_{max}$ used in initialization and training for parameters $\Delta$ (in baselines) and $\xi$ (S4D-DFouT), respectively. In particular, the results show that the S4D-DfouT initialization scheme is less sensitive to the $\xi$ parameter compared to the baselines, especially on the datasets that present a strong frequential component (`Image` and `Pathfinder`).

Next, we ablate several discrete-domain initialization variants. First, we assign each pole a random phase by sampling its imaginary component as $\Omega_n \sim \mathcal{U}[0, 2\pi)$, called S4D-RndImag. Next, we evaluate a token synchronization scheme, which aligns pole angles to periodic intervals every *k*-tokens. Finally, we try a different S4D-DFouT synchronization using batches (details in Appendix C.6).

Table 2: Accuracy on the ablation experiments, averaged over three random seeds (std. in parentheses).

| Method | sCIFAR | SC35 | |
|---|---|---|---|
| | | 16kHz | 8kHz |
| S4-LegS | 91.80 (0.43) | 96.08 (0.15) | 91.32 (0.17) |
| S4-FouT | 91.22 (0.25) | 95.27 (0.20) | 91.59 (0.23) |
| S5 | 90.10 (-) | 96.53 (-) | 94.53 (-) |
| S4D-LegS | 89.92 (1.69) | 95.83 (0.14) | 91.08 (0.16) |
| S4D-Inv | **90.69** (0.06) | 96.18 (0.27) | 91.80 (0.24) |
| S4D-Lin | 90.42 (0.03) | **96.25** (0.03) | 91.58 (0.33) |
| S4D-DFouT | 89.87 (0.07) | 96.07 (0.19) | **93.85** (0.27) |
| S4D-Token | 84.29 (0.14) | 94.93 (0.17) | 92.92 (0.22) |
| S4D-RndImag | 85.68 (0.27) | 95.43 (0.26) | 91.06 (0.47) |
| S4D-Batched-DFouT | 88.67 (0.11) | 95.31 (0.09) | 0.07 (0.14) |

## 6 Conclusion and Limitations

Although LRA tasks are considered hard in the literature, especially tasks as PathX that are designed to stress a model's long-range reasoning, our analysis reveals that SSMs often **succeed in it by exploiting local biases** rather than learning truly global kernels. As a result, even the **hardest LRA benchmarks can be solved with receptive fields much shorter than the input length** as long as the task contains principal frequency components that could be captured by the SSM's spectral modes. Prior initializations in the continuous domain hedge against unknown timescales by

sampling $\Delta$ over a wide range. In this work, we show that our initialization in the discrete domain achieves uniform, alias-free spectral support, eliminating the need for $\Delta$ tuning. These **insights indicate that the intrinsic difficulty of LRA tasks may be overestimated**, underscoring **the need for new benchmarks** that cannot be circumvented through the presented learning biases and that truly challenge models to capture long-range dependencies.

We note that S4D-DFouT initialization did not help in solving a more challenging problem of permuted `psCIFAR`, achieving 65.7% accuracy (see Appendix C.7). Furthermore, we did not explore our proposed initialization in the context of textual data and LLMs, which we leave for a future study.

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

# A  Theoretical insights

## A.1  Transfer Function and Stability Conditions

Consider a continuous-time diagonal state-space system

$$\frac{d}{dt}h(t) = \Lambda h(t) + Bx(t), \quad y(t) = C^\top h(t), \quad t > 0,$$

and its corresponding discretized model (for the sake of simplicity, we do not enforce the real output as in (1))

$$\Sigma : \quad \begin{aligned} h[l+1] &= \overline{\Lambda}\, h[l] + \overline{B}\, x[l]\,, \\ y[l] &= C\, h[l]\,, \end{aligned} \tag{12}$$

where

$$\Lambda = \mathrm{diag}(\lambda_1, \ldots, \lambda_N), \quad \overline{\Lambda} = \mathrm{diag}(\overline{\lambda}_1, \ldots, \overline{\lambda}_N),$$

$$B = \begin{bmatrix} B_1 & \cdots & B_N \end{bmatrix}^\top \quad \overline{B} = \begin{bmatrix} \overline{B}_1 & \cdots & \overline{B}_N \end{bmatrix}^\top,$$

$$C = \begin{bmatrix} C_1 & \cdots & C_N \end{bmatrix}.$$

Focusing on the discretized model, the transfer function in the $Z$–domain, by definition, is given by

$$H(z) = Y(z)/U(z) = C\,(zI - \overline{\Lambda})^{-1}\overline{B}.$$

Because $\overline{\Lambda}$ is diagonal, this simplifies entry-wise to a sum of first-order terms

$$H(z) = \sum_{n=1}^{N} \frac{C_n\,\overline{B}_n}{z - \overline{\lambda}_n} = \sum_{n=1}^{N} \frac{C_n\,\overline{B}_n\, z^{-1}}{1 - \overline{\lambda}_n\, z^{-1}}.$$

The poles of $H(z)$ correspond to the eigenvalues of $\overline{\Lambda}$, $z = \{\overline{\lambda}_n\}_{n=1}^{N}$. The discrete-time SSM is stable if and only if all poles lie inside the unit circle:

$$|\overline{\lambda}_n| < 1, \quad n = 1, 2, \ldots, N$$

**Direct pole training.** It is possible for diagonal SSMs models whose poles correspond to trainable parameters to control them to satisfy stability conditions directly. If the continuous diagonal system is stable, i.e. $\Re(\lambda_n) < 0$, then under the zero-order hold discretization with step $\Delta \in \mathbb{R}^+$, the magnitude of the poles become

$$|\overline{\lambda}_n| = |e^{\Delta\lambda_n}| = |e^{\Delta\Re(\lambda_n)}| < 1. \tag{13}$$

Thus, the discretized SSM remains stable.

## A.2  $\mathcal{H}_\infty$ Score

In robust control [31], the $\mathcal{H}_\infty$ *score* of a discrete-time LTI system, such as $\Sigma$ in (12), with the transfer function matrix $H$, is defined as

$$\mathcal{H}_\infty = \mathcal{H}_\infty(\Sigma) := \|H\|_\infty^2 := \sup_{\theta \in [0, 2\pi]} \sigma\big(H(e^{j\theta})\big)^2,$$

where $\sigma(\cdot)$ denotes the maximum singular value of a matrix. This score captures the worst-case amplification of disturbances across all frequencies. Crucially, it also provides an energy bound

$$\|y\|_2^2 \leq \|H\|_\infty \|x\|_2^2,$$

so that the total output energy can never exceed the input energy scaled by $\mathcal{H}_\infty$.

For a diagonal discrete SSM in (12), each scalar-state subsystem $\Sigma_i$ has transfer function

$$H_n(z) = \frac{C_n\,\overline{B}_n}{z - \overline{\lambda}_n}$$

hence a closed-form $\mathcal{H}_\infty$ score

$$\mathcal{H}_\infty(\Sigma_n) = \frac{\|C_n\|^2\,\|B_n\|^2}{(1 - |\lambda_n|)^2}$$

highlighting how pole magnitudes limit worst-case gain. This score directly measures each state's worst-case contribution to output energy. This score underpins modal-truncation-style pruning [16].

# B  Proofs

**Proposition 1.** *Let the S4D-Lin convolutional kernel under ZOH discretization be $\overline{K}[l] = 2\,\Re(\sum_{n=0}^{N-1} C_n\,\overline{\lambda}_n^{\ell})$. If we choose $\Delta = \frac{2}{\tau}$, where $\tau \in \mathbb{Z}_{>0}$, and take $C_i = 1$ for all $i$, then $\overline{K}$ presents a "spike" at $l = \tau$, with $|\overline{K}[l]| < |\overline{K}[\tau]|$ ($\forall l \neq \tau$). Moreover, if we take $C$ to be a trainable parameter in general, the Vandermonde matrix $(V_{ln}) = (\overline{\lambda}_n^{l})$ is well-conditioned and gradient descent for $C$ converges at a linear rate.*

*Proof.* We proceed in five detailed steps:

**1. Discrete poles via ZOH.** For the sake of simplicity, let's suppose poles of the continuous system have no decay $w_n = i\pi n$; then, ZOH gives

$$d_n = \exp(\Omega) = \exp(w_n\,\Delta) = \exp(i\pi n\Delta).$$

**2. Fundamental discrete period.** The first harmonic $n = 1$ has frequency $\Omega_1 = \pi\Delta$. A full $2\pi$ rotation requires $\Omega_1\,T = 2\pi \implies T = 2/\Delta$. Thus one cycle of $n = 1$ takes precisely $T = 2/\Delta$ steps.

**3. Matching $T$ to $\tau$.** Set $T = \tau$, i.e. $\Delta = 2/\tau$. Then for any $n$,

$$d_n^\tau = \exp\bigl(i\,n\,\pi\,\Delta\,\tau\bigr) = \exp(i\,2\pi\,n) = 1,$$

so *all* modes align in phase at step $l = \tau$.

**4. Kernel evaluation.** The SSM kernel is

$$K[l] = \Re\Bigl(2\sum_{n=0}^{N-1} C_n\,d_n^l\Bigr) = \Re\Bigl(2\sum_{n=0}^{N-1} d_n^l\Bigr).$$

At $l = \tau$, since $d_n^\tau = 1$ for all $n$,

$$K[\tau] = 2\,\Re\Bigl(\sum_{n=0}^{N-1} 1\Bigr) = 2N,$$

whereas for $l \neq \tau$, the $d_n^l$ are not all unity and partially cancel, yielding $|K[l]| < K[\tau]$.

**5. Conditioning and GD.** Fix an integer delay $\tau > N$. Then for $0 \leq l \leq \tau$ and $0 \leq n \leq N - 1$, the Vandermonde matrix

$$V_{l,n} = d_n^l = \exp\Bigl(i\,\frac{2\pi n}{\tau}\,l\Bigr)$$

is exactly a $(\tau + 1) \times N$ partial Fourier matrix. Concretely, its Gram matrix is

$$V^*V = \Bigl[\sum_{l=0}^{\tau} d_n^l\,\overline{d_m^l}\Bigr]_{n,m} = \sum_{l=0}^{\tau} e^{i2\pi(n-m)l/\tau} = \begin{cases} \tau + 1, & n = m, \\ 0, & n \neq m \end{cases},$$

so $V^*V = (\tau + 1)\,I_N$, where the latter is the identity matrix. Hence, the eigenvalues of $V^*V$ are all equal

$$\lambda_{\min}(V^*V) = \lambda_{\max}(V^*V) = \tau + 1,$$

and the condition number of $V$ is exactly one.

Now consider fitting the target vector $y \in \mathbb{R}^{\tau+1}$ by minimizing $\mathcal{L} = \frac{1}{2}\|VC - y\|^2$ over $C \in \mathbb{C}^N$. Since the Hessian is $\nabla^2\mathcal{L}(C) = V^*V = (\tau + 1)I$, we have that every eigenvalue of it is exactly $(\tau + 1)$, so $\mathcal{L}$ is $(\tau + 1)$-*strongly convex*. Furthermore, the largest eigenvalue (smoothness constant) is $\tau + 1$, thus the function is also $(\tau + 1)$-*smooth*. Consequently, the optimal gradient-descent step-size

$$\eta = \frac{1}{\lambda_{max}(V^*V)} = \frac{1}{\tau + 1},$$

reaches the exact solution in one step.

At each iteration $t$, gradient descent update $C^{(t+1)} = C^{(t)} - \eta \nabla \mathcal{L}(C^{(t)})$, and for the least-squares loss $\mathcal{L}(C)$, the gradient is

$$\nabla \mathcal{L}(C) = \frac{\partial}{\partial C} \frac{1}{2} (VC - y)^* (VC - y) = V^*(VC - y).$$

Therefore, choosing $\eta = \dfrac{1}{\tau + 1}$ and using $V^*V = (\tau + 1)I$, we have

$$C^{(1)} = C^{(0)} - \eta V^*(VC^{(0)} - y)$$
$$= C^{(0)} - \frac{1}{\tau + 1} V^*(VC^{(0)} - y)$$
$$= y \frac{V^*}{\tau + 1} = \frac{y}{V} = C^*$$

More generally any $0 < \eta < \frac{2}{\tau+1}$ still guarantees linear convergence at rate $\|C^{(t)} - C^*\| \leq (1 - \eta(\tau + 1))^t \|C^{(0)} - C^*\|$.

This completes the proof. □ □

## C   Experimental details

In this section, we provide descriptions of the datasets, task formulations, model architectures, and training protocols used throughout our evaluations.

### C.1   Continuous Copying Task

The Continuous Copying (or `Delay`) task is posed as a sequence-to-sequence problem in which the model must reproduce the input delayed by $\tau = 1000$ time steps. Specifically, we utilize white-noise sequences of length 4000 samples, band-limited to 1000 Hz, and train a single linear state-space model with state dimension $N = 1024$ end-to-end for 20 epochs.

### C.2   Pixel-level image classification

The serialized CIFAR-10 (`sCIFAR`) dataset is constructed by flattening the $32 \times 32$ color images into a one-dimensional sequence of length 1024. All sequences are then standardized to zero mean and unit variance. In the permuted serialized (`psCIFAR`) variant, the same preprocessing is applied, but with a single fixed random permutation of the 1024 pixel positions before flattening, thereby removing any local spatial structure. The permutation vector utilized is included in the supplementary code. Ablation results on `psCIFAR` are available in Table 3.

### C.3   Long Range Arena

The Long Range Arena (LRA) [17] benchmark comprises seven diverse sequence-classification tasks designed to test a model's ability to capture long-range dependencies:

**ListOps.** The dataset consists of mathematical expressions formed by nested operators—such as `min` and `max`—applied to integer numbers in the range 0–9, and written in prefix notation with explicit brackets. Each character is represented as a one-hot vector over 17 possible tokens (with all opening brackets and operators grouped into a single token). Because sequence lengths vary, shorter sequences are padded with a special padding symbol up to a maximum length of 2048. The task is to classify each expression into one of 10 classes, corresponding to its integer result.

**Text.** The IMDb sentiment classification task is as follows, given a movie review represented as a sequence of character tokens, the goal is to predict whether the review is positive or negative. Each character is one-hot encoded over 129 possible tokens. Because reviews vary in length, sequences are padded to a maximum length of 4096.

Table 3: Results on additional datasets. Accuracy for `psCIFAR` and RMSE for predicting respiratory rate (RR), heart rate (HR), and blood oxygen (SpO$_2$).

| Method | psCIFAR Acc. ($\uparrow$) | BIDMC-HR RMSE ($\downarrow$) | BIDMC-RR RMSE ($\downarrow$) | BIDMC-SpO$_2$ RMSE ($\downarrow$) |
|---|---|---|---|---|
| S4-LegS | 64.96 | 0.332 | 0.247 | 0.090 |
| S4-FouT | 65.27 | 0.339 | 0.301 | 0.068 |
| S4D-LegS | - | **0.367** | 0.248 | 0.102 |
| S4D-Lin | 64.51 | 0.379 | 0.226 | 0.114 |
| S4D-Inv | 64.88 | 0.373 | 0.254 | 0.110 |
| S4D-DFouT | **65.72** | 0.415 | 0.273 | 0.122 |
| S4D-Token | 60.25 | 0.490 | 0.440 | 0.126 |
| S4D-RndImag | 60.55 | 0.437 | **0.216** | **0.087** |
| S4D-Batched-DFouT | 63.61 | 0.456 | 0.344 | 0.118 |

**Retrieval.** The task is based on the ACL Anthology network corpus; given two textual citations encoded as sequences of character tokens, predict whether they refer to the same publication. Each character is one-hot encoded over 97 possible tokens. Sequences vary in length and are padded to a maximum of 4000 tokens using a special padding symbol, and a reserved end-of-sequence token is appended. The model outputs one of two classes: equivalent or non-equivalent.

**Image.** Considers the CIFAR-10 image classification task, but images are first converted to a single-channel grayscale intensity map, and pixel values are normalized to zero mean and unit variance across the entire dataset. The resulting $32 \times 32$ image is flattened into a sequence of length 1024. All sequences are of equal length, and the model predicts one of ten classes corresponding to the 10 categories.

**Pathfinder.** In this task, the images contain two marked points (start and end) represented by small circles, and a set of dashed line segments. The objective is to predict whether there exists a continuous dashed-line path connecting the start and end points. Each input is a $32 \times 32$ grayscale image with pixel values normalized to the range $[-1, 1]$.

**PathX**. An "extreme" version of the Pathfinder challenge. Instead, the images are $128 \times 128$ pixels in the `PathX-128`, or 256 x 256 in the `PathX-256`, resulting in much longer sequences. As denoted in Table1 by (†), we manage to successfully train S4D-Lin on PathX by increasing the range from $(\Delta_{min}, \Delta_{max}) = (0.0001, 0.01)$ to $(\Delta_{min}, \Delta_{max}) = (0.0001, 0.1)$. This is done in order to accommodate the fundamental harmonic within the model frequencies explored at initialization. This requires $\Delta \geq 2/128 = 0.015$, which was not covered in the initial range proposed in [12].

## C.4   Speech Commands

We use the full 35-class Speech Commands dataset [29], which comprises 1 second of recorded voice commands sampled at 16 kHz. Additionally, as introduced in [12], we also test the zero-shot performance when the signal is undersampled at 8kHz.

## C.5   BIDMC Vital signs

The BIDMC dataset [32] consists of continuous physiological signals from raw EKG and PPG traces: heart rate (HR), respiratory rate (RR), and blood oxygen levels (SpO$_2$) over length-4000 signal,s which are normalized per channel to zero mean/unit variance.

## C.6   Ablations details

We compare three variants of discrete-domain initialization beyond the main S4D-DFouT scheme. The ablation methods share the real part with DFouT, i.e. $\exp(-\xi/2)$, while the imaginary part is configured as follows:

- **S4D-Token:** Discrete frequencies are chosen to resonate at integer periods $n = 1, 2, \ldots, N$, i.e. $\Omega_n = 2\pi/n$. This assigns one pole per token in the sequence, yielding a coarse-grained coverage of the frequency spectrum biased towards low frequencies.

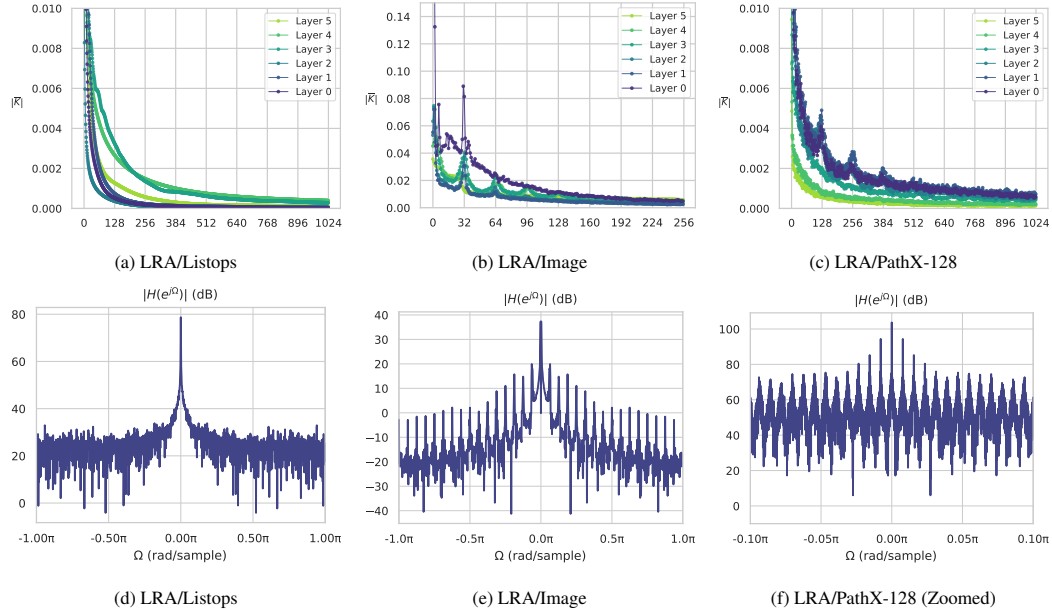

Figure 7: Subplots (a)–(c) show the mean absolute value of the learned discrete kernel $|\overline{K}[l]|$ for each of the six SSM layers on three LRA benchmarks—(a) `ListOps`, (b) `Image`, and (c) `PathX-128`—where only the first $L'$ coefficients are displayed on the x-axis to facilitate the visualization. Subplots (d)–(e) present the magnitude of the discrete Fourier transform (in dB) for $10\,000$ randomly sampled sequences from the same datasets. These visualizations demonstrate that datasets with strong local structure—such as the serialized image benchmarks `sCIFAR` and `PathX-128`—exhibit clear frequency components corresponding to row-wise correlations. This is reflected in the learned kernels to exploit a "local attention" profile. In `sCIFAR`, a pronounced peak in the kernel occurs at multiples of 32-pixel (the row-stride), which corresponds to a clear frequency component ($\Omega = 2\pi/32$); whereas `PathX-128` shows periodic modes at 128 pixels, i.e., frequencies of $\Omega = 2\pi/128$ along with their higher-frequency harmonics. By contrast, the `ListOps` benchmark exhibits no such localized frequency peaks, reflecting its lack of inherent local structure.

- **S4D-RndImag:** Each imaginary component $\Omega_n$ is i.i.d. sampled from the uniform distribution $\mathcal{U}[0, 2\pi)$.

- **S4D-Batched-DFouT:** Synchronization across the $H$ SSMs in each layer is achieved by partitioning the spectrum into $H$ contiguous blocks, each of size $N$. Therefore, the $h$-th SSM is assigned with a batch of $N$ adjacent frequencies shifted by a phase $\phi_h \in [0, 2\pi)$:

$$\textbf{Batched-DFouT: } \overline{\lambda}_{h,n} = \exp\left(-\frac{\xi_h}{2} + i\left(\frac{2\,\pi\,n}{NH} + \phi_h\right)\right) \text{ where } \phi_h = \frac{2\,\pi\,(h-1)}{H}. \qquad (14)$$

### C.7 Extended Results

In Figure 7 we show the kernel visualization together with the discrete Fourier transform on three representative LRA benchmarks. These visualizations demonstrate that datasets with strong local structure—such as the serialized image benchmarks `sCIFAR` and `PathX`—exhibit clear frequency components corresponding to row-wise correlations. SSM models exploit this fact to learn kernels showing a "local attention" profile. The learned kernels "attends" primarily to its nearby inputs, imposing an inductive bias toward locality. By contrast, the `ListOps` benchmark exhibits no such localized frequency peaks, reflecting its lack of inherent local structure.

Figure 8 extends Figure 6 from the main manuscript to the remaining datasets LRA benchmarks. Here, we perform an ablation study on the effect of upper bounds $\Delta_{max}$ and $\xi_{max}$ used in initialization and training for parameters $\Delta$ (in baselines) and $\xi$ (S4D-DFouT), respectively. The results show that the S4D-DFouT initialization scheme is less sensitive to the $\xi$ parameter compared to the baselines, especially on the dataset which present a strong frequential component (`Image` and `Pathfinder`).

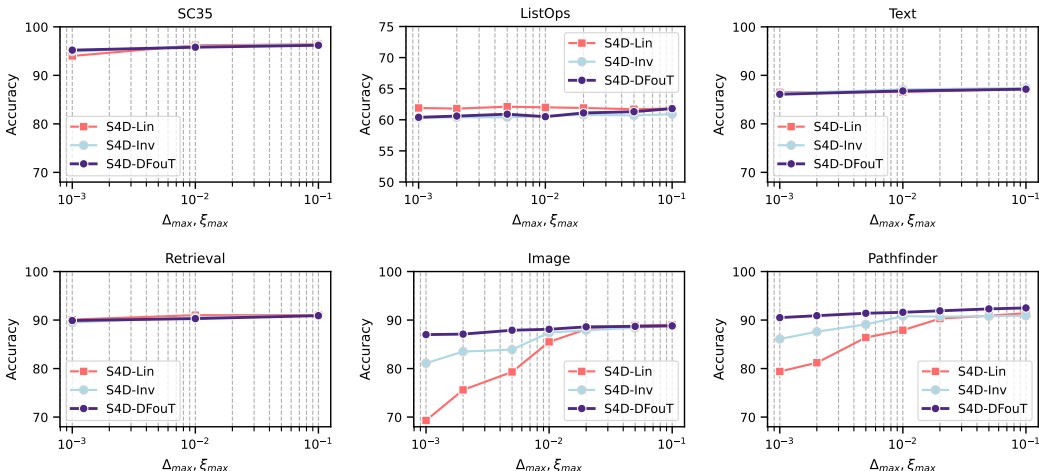

Figure 8: Ablation experiment in LRA. Accuracy upon different initialization schemes for fixed $\Delta_{min}, \xi_{min} = 10^{-3}$ and variable $\Delta_{max}, \xi_{max}$. Datasets `Image` and `Pathfinder` presenting a strong frequential component are highly sensitive to an initialization that captures that frequency.

Finally, an extended comparison on Long Range Arena is presented in Table 4. It spans across all seven tasks, including the most challenging `PathX-256` setting. While prior work denoted as "S4 + Self-Pretrain" [28] (marked †) also succeed on it, it does relying on self-pretraining. Whereas our S4D-DFouT achieves it training from scratch.

Table 4: Extended comparison on LRA. (†) denotes the method requires self-pretraining.

| Model | ListOps (2,048) | Text (4,096) | Retrieval (4,000) | Image (1,024) | Pathfinder (1,024) | PathX-128 (16,384) | PathX-256 (65,536) |
|---|---|---|---|---|---|---|---|
| Transformer [26] | 36.37 | 64.27 | 57.46 | 42.44 | 71.40 | ✗ | ☐ |
| Reformer [33] | 37.27 | 56.10 | 53.40 | 38.07 | 68.50 | ✗ | ☐ |
| BigBird [34] | 36.05 | 64.02 | 59.29 | 40.83 | 74.87 | ✗ | ☐ |
| Linear Trans. [35] | 16.13 | 65.90 | 53.09 | 42.34 | 75.30 | ✗ | ☐ |
| Performer [36] | 18.01 | 65.40 | 53.82 | 42.77 | 77.05 | ✗ | ☐ |
| MEGA [25] | 63.14 | 90.43 | 91.25 | 90.44 | 96.01 | 97.98 | ☐ |
| DSS [11] | 60.60 | 84.80 | 87.80 | 85.70 | 84.60 | 87.80 | ☐ |
| S4D-LegS [12] | 60.47 (0.34) | 86.18 (0.43) | 89.46 (0.14) | 88.19 (0.26) | 93.06 (1.24) | 91.95 | ☐ |
| S4D-Inv [12] | 60.18 (0.35) | 87.34 (0.20) | 91.09 (0.01) | 87.83 (0.37) | 93.78 (0.25) | 92.80 | ☐ |
| S4D-Lin [12] | 60.52 (0.51) | 86.97 (0.23) | 90.96 (0.09) | 87.93 (0.34) | 93.96 (0.60) | ✗ | ☐ |
| S4-FouT [9] | 57.88 (1.90) | 86.34 (0.31) | 89.66 (0.88) | 89.07 (0.19) | 94.46 (0.24) | ✗ | ☐ |
| S4-LegS [9] | 59.60 (0.07) | 86.82 (0.13) | 90.90 (0.15) | 88.65 (0.23) | 94.20 (0.25) | 96.35 | ☐ |
| Liquid-S4 [23] | 62.75 (0.20) | 89.02 (0.04) | 91.20 (0.01) | 89.50 (0.40) | 94.80 (0.20) | 96.66 (0.001) | ☐ |
| S5-Inv [14] | 60.07 (0.26) | 87.77 (0.29) | 91.26 (0.12) | 86.41 (0.17) | 93.42 (0.42) | 97.54 (0.74) | ☐ |
| S5-Lin [14] | 59.98 (0.53) | 88.15 (0.24) | 91.31 (0.24) | 86.05 (0.96) | 94.31 (0.36) | 65.60 (27.00) | ☐ |
| S5 [14] | 62.15 (0.23) | 89.31 (0.15) | 91.40 (0.05) | 88.00 (0.22) | 95.33 (0.26) | 98.58 (0.17) | ☐ |
| LRU [13] | 60.2(0.8) | 89.4(0.1) | 89.9(0.1) | 89.0(0.1) | 95.1(0.1) | 94.2(0.4) | ☐ |
| S4 + Self-Pretrain [28](†) | 61.25 | 90.34 | 88.74 | 89.36 | 94.92 | 96.94 | 87.11 |
| S4D-DFouT | 61.89 | 87.41 | 90.95 | 88.48 | 94.30 | 94.17 | 87.89 |

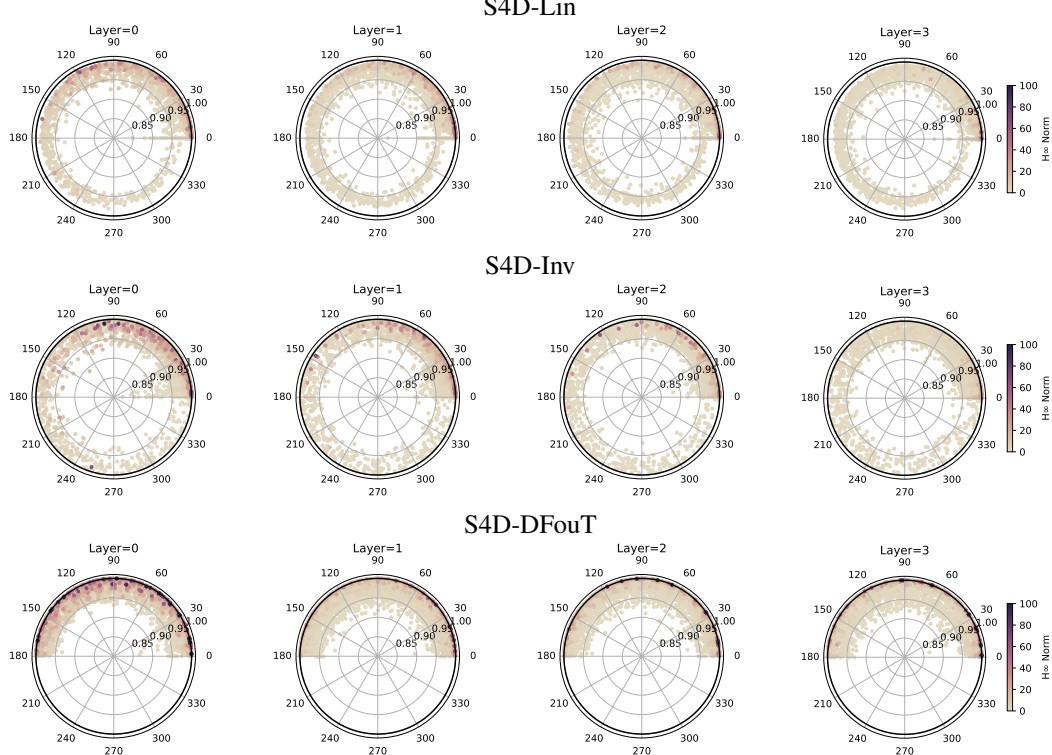

Figure 9: Polar plots of the learned complex poles on `sCIFAR` for three initialization schemes—S4D-Lin (top row), S4D-Inv (middle row), and S4D-DFouT (bottom row). Poles are shaded by their $\mathcal{H}_\infty$-norm (warmer tones indicate higher norms), emphasizing the modes that drive the system's worst-case gain. The dashed unit circle marks the stability boundary. S4D-Lin entanglement between decay and frequency can be observed, resulting in spiral-like clustering of poles; S4D-Inv spreads poles more evenly around the circle; and S4D-DFouT maintains the original half-plane distribution of poles throughout training.

## C.8 Hyperparameters

The S4D-DFouT hyperparameter configuration we adopt in the experimentation is provided in Table 5.

Table 5: Hyperparameters used for the S4D-DFouT reported results. L denotes the number of layers; H, the embedding size; N, the hidden dimension; Dropout, the dropout rate; Lr, the global learning rate; Bs, the batch size; Epochs, the maximum number of training epochs; WD, weight decay; and $(\xi_{\min}, \xi_{\max})$, the range of decay rate values.

| Task | L | H | N | Norm | Pre-norm | Dropout | Lr | Bs | Epoch | Wd | $(\xi_{\min}, \xi_{\max})$ |
|---|---|---|---|---|---|---|---|---|---|---|---|
| sCIFAR | 6 | 128 | 64 | LN | False | 0.1 | 0.01 | 64 | 100 | 0.05 | (0.001, 0.1) |
| psCIFAR | 6 | 128 | 64 | LN | False | 0.1 | 0.01 | 64 | 100 | 0.05 | (0.001, 0.1) |
| ListOps | 6 | 256 | 64 | BN | False | 0 | 0.01 | 50 | 40 | 0.05 | (0.001, 0.1) |
| Text | 6 | 256 | 64 | BN | True | 0 | 0.01 | 16 | 32 | 0.05 | (0.001, 0.1) |
| Retrieval | 6 | 256 | 64 | BN | True | 0 | 0.01 | 64 | 20 | 0.05 | (0.001, 0.1) |
| Image | 6 | 512 | 64 | LN | False | 0.1 | 0.01 | 50 | 200 | 0.05 | (0.001, 0.1) |
| Pathfinder | 6 | 256 | 128 | BN | True | 0 | 0.001 | 64 | 200 | 0.03 | (0.001, 0.1) |
| PathX-128 | 6 | 256 | 128 | BN | True | 0 | 0.0005 | 32 | 50 | 0.05 | (0.0001, 0.1) |
| PathX-256 | 6 | 256 | 128 | BN | True | 0 | 0.0001 | 16 | 20 | 0.05 | (0.0001, 0.1) |
| SC35 | 6 | 128 | 64 | BN | True | 0 | 0.001 | 16 | 40 | 0.05 | (0.001, 0.1) |
| BIDMC | 6 | 128 | 256 | BN | True | 0 | 0.01 | 32 | 500 | 0.05 | (0.001, 0.1) |

