# OpenReview forum: "Uncovering the Spectral Bias in Diagonal State Space Models"
_NeurIPS.cc/2025/Conference — NeurIPS 2025 poster_

### Official Review · Reviewer_bKH5 · 2025-06-27

**Clarity:** 2
**Significance:** 3
**Originality:** 3
**Rating:** 4
**Confidence:** 3

**Summary:**

In this paper, the authors study the initialization of diagonal SSMs from a frequency perspective. They observe that in previous work, the diagonal state matrix is first initialized in the continuous domain and then discretized using a learnable discretization parameter during training. This method has several disadvantages, including the coupling between the decay rate and the oscillation frequency through the discretization parameter, a lack of interpretability or fine-grained control, and misalignment with the needs of real-world tasks. To address these limitations, the authors propose a universal initialization scheme for discrete SSMs from a frequency perspective, where all poles are placed uniformly around the unit circle in the complex plane and modulated by a shared exponential decay. Unlike previous methods, the learnable discretization parameter is removed, thereby eliminating the entanglement between decay and frequency components. Experimental results demonstrate that the proposed initialization scheme enables SSMs to achieve superior performance on the Long Range Arena benchmark. Additionally, it facilitates training SSMs on extremely large datasets, which was previously challenging.

**Questions:**

1. It seems that there is some typos in Eq. (8). The correct form should be $$\sum _{n=0}^{N-1} C_n \overline{B} _n \sum _{l \geq 0} e^{-(\Delta \alpha _n + i(\theta - \Omega_n)l)} = \sum _{n=0}^{N-1} \frac{C_n \overline{B} _n}{1-e^{-\Delta \alpha _n} e^{i (\theta - \Omega_n)}}.$$
2. In the proof of Proposition 1, the definition of $d _n$ seems to be confusing. At step 1, $d _n$ is defined as $d _n = e^{-\frac{\Delta}{2}} e^{i \pi n \Delta}$. Thus, we have $d _n^k = e^{-\frac{k \Delta}{2}} e^{i k \pi n \Delta}$. Further, if $\Delta = 2/\tau$, we can derive that $d _n^k = e^{-\frac{k}{\tau}} e^{\frac{2 i k \pi n}{\tau}}$. However, the authors show that $d _n^k = e^{\frac{2 i k \pi n}{\tau}}$ holds in step 5. Can the authors provide more explanations on this?

**Ethical Concerns:**

["NO or VERY MINOR ethics concerns only"]

**Final Justification:**

The authors have adequately addressed my concerns in the rebuttal. There do exist some typos in Equations or Theorems, yet the authors have fully corrected them in the rebuttal, and I also believe that they do not affect the final results. From my personal perspective, this work is interesting and technical solid, and I think that it meets the conference's acceptance criteria.

**Limitations:**

The study's limitations are explicitly addressed in the final section of the main text.

**Paper Formatting Concerns:**

I do not find any formatting issues in this paper.

**Quality:**

3

**Strengths And Weaknesses:**

**Strengths**
- The motivation of this work is clear and reasonable. From a frequency perspective, the authors propose a more natural and general initialization scheme for discrete SSMs. The experimental results are comprehensive and sufficient to support the effectiveness of the proposed method.
- From my personal perspective, the proposed initialization scheme is novel. The inspiration comes from the spectral domain, which is different from previous initialization approaches that are designed in the spatial domain.

**Weakness**
- Theoretical analysis of why the proposed initialization scheme enables SSMs to achieve better performance remains limited. In Proposition 1, the authors demonstrate that a previous method, S4D-Lin, fails to solve the continuous copying task when the discretization parameter is selected inappropriately. However, the authors do not provide theoretical analysis of the learning dynamics of SSMs under their proposed initialization. As a result, it remains unclear why their method enables SSMs to converge to better optima and achieve improved generalization performance.
- The presentation quality could be enhanced by correcting typographical errors and providing more detailed explanations. Specific suggestions are provided in the Questions section.

---

> ### Author Rebuttal · Authors · 2025-07-31
>
> Dear reviewer thank for your time in assessing our paper, and for the valuable comments.
>
> > **Reviewer[bKH5] Point #1**: "Theoretical analysis of why the proposed initialization scheme enables SSMs to achieve better performance remains limited. In Proposition 1, the authors demonstrate that a previous method, S4D-Lin, fails to solve the continuous copying task when the discretization parameter is selected inappropriately. However, the authors do not provide a theoretical analysis of the learning dynamics of SSMs under their proposed initialization. As a result, it remains unclear why their method enables SSMs to converge to better optima and achieve improved generalization performance."
>
> In fact, as shown in the center panel of Figure 4, our method actually performs marginally worse than S4D‑Lin under the optimal configuration of $\Delta$, which, as demonstrated in Proposition 1, converges to the optimal solution for S4D-Lin. The advantage we want to highlight in that motivating experiment on the Continuous Copying task is that our initialization S4D-DFouT does not require tuning of $\Delta$, achieving competitive results where previous initializations are severely sensitive to the selection of $\Delta$ hyperparameter.
>
> To really motivate with a real example the hardness of selecting $\Delta$, we also discuss in Section 5.3 the following case. In the literature, S4D-Lin has repeatedly underperformed, failing to solve challenges like PathX-128 that other initializations could handle. We show that this shortfall is not an intrinsic limitation of the linear initialization, but rather of the wrong selection of the $\Delta$ hyperparameter in the literature. To capture a period of 128, according to our insights, we require $\Delta > 2/128 \approx 0.015$; however in the literature [2] a range of [0.0001, 0.01] was used, just by increasing the range over the calculated threashold we suceed in learning the task (marked with "†" in Table 1).
>
>
> > **Reviewer[bKH5] Question #1**: Typos and erratas. It seems that there is a typo in Eq.(8). In the proof of Proposition 1, the definition of $d_n$ seems to be confusing.
>
> - The typo on the sign on Eq.(8) will be corrected.
>
> - Thank you for pointing this mistake in the Proposition. We corrected the statement to emphasize the assumption that is no decay in the poles. Therefore, the Proposition 1 is as follows:
>
> **Proposition 1.**
> Let the S4D‑Lin convolutional kernel under zero-order hold (ZOH) discretization be defined as
> $\overline{K}[l] = 2\mathrm{Re} \left( \sum_{n=0}^{N-1} C_n \, \overline{\lambda}_n^\ell \right)$,  where $\overline{\lambda}_n$ are the discrete-time eigenvalues and $C_n \in \mathbb{C}$.
>
> If we choose $\Delta = \frac{2}{\tau}$ and $\alpha = 0$ (**no decay in the poles**), for some $\tau \in \mathbb{Z}_{>0}$, and set $C_n = 1$ for all $n$,  then $\overline{K}$ exhibits a spike at $l = \tau$, satisfying  $|\overline{K}[l]| < |\overline{K}[\tau]|$ for all $l \ne \tau$.
>
> Moreover, if we treat $C = (C_0, \dots, C_{N-1})$ as trainable parameters,  then the associated Vandermonde matrix defined by $V_{\ell n} = \overline{\lambda}_n^\ell$  is well-conditioned, and gradient descent on $C$ converges at a linear rate.

---

### Official Review · Reviewer_eFzT · 2025-06-29

**Clarity:** 4
**Significance:** 1
**Originality:** 2
**Rating:** 3
**Confidence:** 4

**Summary:**

This paper investigates new initialization schemes for diagonal state space models (SSMs) to address two issues: first, the existing methods often mix frequency and decay, making the models too sensitive to the chosen time step during discretization; second, the frequency coverage of these methods doesn't usually match the needs of real-world tasks.
To overcome these problems, the authors introduce a new initialization scheme. The authors test their method on several benchmarks, including synthetic tasks like the continuous copying task, and real-world task like speech classification, and the Long Range Arena benchmark.

**Questions:**

While it may be outside the current scope of the paper, I’m curious about how the state matrix weights evolve throughout training. Analyzing this could provide further insight into why the proposed initialization leads to better performance, and whether its benefits persist or diminish as training progresses.

**Ethical Concerns:**

["NO or VERY MINOR ethics concerns only"]

**Final Justification:**

Writing the same justification I wrote to the Authors:

While this assessment is inherently subjective, I maintain my stance that the scope of the paper is still too narrow for acceptance into the conference. The primary reason is that the subject matter (initialization schemes for discretized state space models (SSMs)) does not hold sufficient significance for two main reasons:

1. As previously highlighted in my original review, simpler models exist today, such as LRU, which are expressively equivalent to S4 but do not require discretization. The authors themselves see in the Appendix that LRU simpler models outperform their newly proposed initialization in approximately 50% of the benchmarks. Specifically, the authors state:

> "Although we included LRU in the results comparison in LRA (Table S4 in Supplementary Material), this work definitely requires better coverage in the related work."

Given these points, I believe initialization schemes of discretized SSMs are not significant enough to justify inclusion in a top-tier conference.

2. An exception to the above critique would have been possible had the empirical evidence demonstrated clear superiority of their method. However, while the authors do show improvements over other S4 initializations, their approach still underperforms against much simpler alternatives such as LRU or newer models like S5.

While I completely understand the difficulties in analyzing selective SSMs like Mamba, due to what is stated above, the importance of the analysis of discretization schemes in non-selective SSMs remains questionable given their complexity and limited performance.

Nevertheless, I acknowledge that the authors' analysis of dynamics and the evolution of learned poles provides valuable insights. In their rebuttal, the authors highlight:

> "We depict in Figure S9 in the Supplementary Material the final distribution of the poles layer-by-layer after training in sCIFAR for the different initialization schemes \[...]. Our major observation is that in all initializations, the angles of the poles (resonant frequencies) remain largely unchanged from their initial values."

This type of analysis indeed contributes meaningfully to the field. However, if the authors feel that this analytical perspective is their primary contribution, then the paper should be positioned as such, with additional experiments to strengthen and expand these insights. As it currently stands, I do not find the contribution substantial enough to recommend acceptance.

*I want to note that while I am very confident in my view, all my reasons for rejecting are listed above. If the Area Chair, with many more years of experience than myself, believes I am setting the bar for acceptance too high, I encourage them to disregard my opinion.*

**Limitations:**

yes.

**Paper Formatting Concerns:**

None.

**Quality:**

2

**Strengths And Weaknesses:**

**Strengths:**

1. The paper is clear, well-written, and easy to follow.
2. It provides a thorough and insightful ablation study that effectively supports its claims.
3. A detailed and honest limitation section.

**Weaknesses:**
1. The paper heavily emphasizes discretization issues in diagonal state space models but does not consider alternative state space methods that avoid discretization entirely, such as LRU (Resurrecting Recurrent Neural Networks for Long Sequences, Orvieto et al.).
2. Due to point 1, discretization concerns are particularly relevant in selective state space models like Mamba (Mamba: Linear-Time Sequence Modeling with Selective State Spaces, Gu and Dao) where discretization is still used, yet the paper overlooks these types of models.

In my opinion, these omissions greatly weaken both the scientific framing and empirical contribution of the paper.

---

> ### Author Rebuttal · Authors · 2025-07-29
>
> Dear reviewer thank for your time in assessing our paper, and for the valuable comments.
>
> > **Reviewer[eFzT] Point #1**: Consider alternative SSMs models: The paper heavily emphasizes discretization issues in diagonal state space models but does not consider alternative state space methods that avoid discretization entirely, such as LRU (Resurrecting Recurrent Neural Networks for Long Sequences, Orvieto et al.).
>
> Thank you for pointing out the reference LRU [4]. Although we included LRU in the results comparison in LRA (Table S4 in Supplementary Material), this work definitely requires better coverage in the related work.
>
> We briefly recall the main findings of the LRU paper. As the Reviewer points,
> LRU does not rely on the discretization of a latent continuous-time system but directly optimizes the magnitude and oscillation frequencies independently. LRU method preserves (and in a sense is motivated by) the Glorot initialization from RNN, which in the limit behavior is de-facto uniform spectral initialization in the complex unit disk.
>
> Yet our contributions are complementary to this work, both in the findings and in the motivation. We first provide a mechanistic interpretation of S4D models from the frequency perspective. Furthermore, we incur on the analysis on other eigenvalue distributions rather than uniform. We show the importance of the frequential coverage aligned with the spectrum of the data and point out some design flaws of previous S4D initializations as the aliasing and Hermitian symmetry. On the practical side, we provide a layer‑by‑layer analysis of what these models learn (Figure 5), observing that initial layers focus explicitly on capturing the frequency present in the data, revealing emergent “attention” patterns in these initial layers. Leveraging these insights, we propose our initialization.
>
> [4] Orvieto, Antonio, et al. "Resurrecting recurrent neural networks for long sequences." International Conference on Machine Learning. PMLR, 2023.
>
> > **Reviewer[eFzT] Point #2**: Selective State Space Models: Discretization concerns are particularly relevant in selective state space models like Mamba (Mamba: Linear-Time Sequence Modeling with Selective State Spaces, Gu and Dao) where discretization is still used, yet the paper overlooks these types of models.
>
>  In Mamba's "selective" state‐space formulation, the initialization of the state matrix A plays a much smaller role than in earlier SSMs architectures. While Mamba does discretize a latent continuous‑time state matrix A, its input‑dependent selection operator $\Delta(x_t)$ dynamically adjusts the effective timestep, making this model less dependent to the initialization scheme. Particularly, in that work [3], authors experiment with three initializations of the SSMs: the real variant is S4D-Real, the complex variant is S4D-Lin, and a random initialization, achieving very similar performance in all cases (see section 4.6.2 of the original work). In Mamba (and more generally in S6 models) the matrix A is not tied to a single fixed discretization step; instead, selectivity produces a context‑dependent state update at every step.
>
> In that non‑stationary domain, providing model interpretability complicates due to the input‑dependent dynamics that evolve at each timestep. Therefore, we limit our work to time-invariant dynamics.
>
>
>
> [3] Gu, Albert, and Tri Dao. "Mamba: Linear-time sequence modeling with selective state spaces." arXiv preprint arXiv:2312.00752 (2023).
>
> > **Reviewer[eFzT] Question #1**: Evolution of poles throughout the training: While it may be outside the current scope of the paper, I’m curious about how the state matrix weights evolve throughout training. Analyzing this could provide further insight into why the proposed initialization leads to better performance, and whether its benefits persist or diminish as training progresses.
>
> Thank you for the question. We depict in Figure S9 in the Supplementary Material the final distribution of the poles layer-by-layer after training in sCIFAR for the different initialization schemes (our proposed and the baselines). Our major observation is that in all initializations, the angles of the poles (resonant frequencies) remain largely unchanged from their initial values. However, poles exhibit a significant change in the decay, presenting a smaller radius. In the particular task of sCIFAR, we believe that some poles are learned to decay rapidly to fine-control the initial coefficients of the kernel, which, as reported in Figure 5, are particularly relevant for performance in this task (especially the first 32 coefficients). Furthermore, the initial distribution is preserved: with S4D-Lin showing the entanglement between decay and frequency,  resulting in spiral-like clustering of the poles; S4D-Inv spreading poles evenly around the circle; and S4D-DFouT maintaining the original half-plane distribution of the poles throughout training.
>
> In the updated manuscript, we will provide an extension of Figure S9, covering the distribution of the poles (with corresponding norms) for different stages throughout the training (current regulations prevent us from sharing figures in the rebuttal).
>
> We appreciate the reviewer’s comments, which helped us clarify both the motivation and broader implications of our work. We believe these revisions better position the paper’s contributions and respectfully invite reevaluation based on the new context.

---

> > ### Comment · Reviewer_eFzT · 2025-08-02
> >
> > I thank the authors for their thoughtful and thorough rebuttal.
> >
> > While this assessment is inherently subjective, I maintain my stance that the scope of the paper is still too narrow for acceptance into the conference. The primary reason is that the subject matter (initialization schemes for discretized state space models (SSMs)) does not hold sufficient significance for two main reasons:
> >
> > 1. As previously highlighted in my original review, simpler models exist today, such as LRU, which are expressively equivalent to S4 but do not require discretization. The authors themselves see in the Appendix that LRU simpler models outperform their newly proposed initialization in approximately 50% of the benchmarks. Specifically, the authors state:
> >
> > > "Although we included LRU in the results comparison in LRA (Table S4 in Supplementary Material), this work definitely requires better coverage in the related work."
> >
> > Given these points, I believe initialization schemes of discretized SSMs are not significant enough to justify inclusion in a top-tier conference.
> >
> > 2. An exception to the above critique would have been possible had the empirical evidence demonstrated clear superiority of their method. However, while the authors do show improvements over other S4 initializations, their approach still underperforms against much simpler alternatives such as LRU or newer models like S5.
> >
> > While I completely understand the difficulties in analyzing selective SSMs like Mamba, due to what is stated above, the importance of the analysis of discretization schemes in non-selective SSMs remains questionable given their complexity and limited performance.
> >
> > Nevertheless, I acknowledge that the authors' analysis of dynamics and the evolution of learned poles provides valuable insights. In their rebuttal, the authors highlight:
> >
> > > "We depict in Figure S9 in the Supplementary Material the final distribution of the poles layer-by-layer after training in sCIFAR for the different initialization schemes \[...]. Our major observation is that in all initializations, the angles of the poles (resonant frequencies) remain largely unchanged from their initial values."
> >
> > This type of analysis indeed contributes meaningfully to the field. However, if the authors feel that this analytical perspective is their primary contribution, then the paper should be positioned as such, with additional experiments to strengthen and expand these insights. As it currently stands, I do not find the contribution substantial enough to recommend acceptance.
> >
> > *I want to note that while I am very confident in my view, all my reasons for rejecting are listed above. If the Area Chair, with many more years of experience than myself, believes I am setting the bar for acceptance too high, I encourage them to disregard my opinion.*

---

> > > ### Author Response · Authors · 2025-08-02
> > >
> > > Dear reviewer : Thank you for your response to our rebuttal. We would like to emphasize three critical points that we hope shed more light on the comparison with LRU method, but also why we strongly believe in the significance of our contributions.
> > >
> > > 1. Our comparison table with LRU method is there for the transparency of our method, but it does not tell the whole story. In particular, when it comes to the performance comparison between LRU and our proposed method, we discuss the Long Range Arena benchmark and sCIFAR classification tasks. In particular, we notice that on the tasks ListOps, Retrieval and sCIFAR, our method yields better performance, while on the tasks Text and Pathfinder LRU performs better.
> > >
> > >    However, we also notice that the size of the state matrices (or the number of poles $N$ per an SSM) in LRU settings for each task is bigger than ours two or  three times at least (for example on Text task, they use N=192, while in our case N=64). We believe that the density of the poles in their initialization is likely covering the poles that we are proposing with our method (and some), hence the perceived performance increase.
> > >
> > > Finally, our contributions go beyond to what LRU paper introduces. In particular:
> > >
> > > 2. We address the problem of underperformance of Fourier based initializations in the literature (S4D-Lin, S5D-Lin), or others such as S4D-Lin and further provide a mechanistic interpretation of how time-invariant SSMs work (analysis of the lobes, spectrum coverage, and kernel length). These insights were crucial in designing our proposed initialization method.
> > >
> > > 3. Our insights as well as the initialization scheme allowed us to be the first to report non-trivial results on the most difficult LRA task - PathX-256.
> > >
> > > Having in mind these points but also the overall content of our paper and the remarked contributions therein, we appreciate your constructive critique, but disagree, respectively, with your given significance score.

---

### Official Review · Reviewer_YT22 · 2025-07-01

**Clarity:** 3
**Significance:** 2
**Originality:** 2
**Rating:** 4
**Confidence:** 4

**Summary:**

This paper presents the S4D-DFouT initialization scheme for deep SSM models. Whereas other methods either use initialization schemes that have no consistent frequency-domain implications, or tie the discrete frequency spectrum to the discretization parameter, DFouT directly targets a well-designed discrete frequency spectrum. Experiments are presented and DFouT-initialized methods appear to perform favorably. Some light interpretability inspections are made.

**Questions:**

1. Doesn’t the locality of the receptive field influence the decay rate as opposed as well the frequency?  Because if I had an energy-preserving eigenvalue then that frequency is repeated indefinitely, meaning its receptive field would become _every_ (e.g.,) 32 pixels?  So why is spacing the frequency components so important, and do you underweight/miss the contribution of amplitude components?

**Ethical Concerns:**

["NO or VERY MINOR ethics concerns only"]

**Final Justification:**

I appreciate the authors comments, particularly in response to W1.  However, the response somewhat validates my core point: it is still essentially an empirical observation that initializing closer to the final distribution yields better performance.  It doesn't then leverage this insight to break new ground or explain computational mechanisms (it just observes this results over and over again).  I think this is fine but it doesn't push me towards campaigning for the acceptance of this paper (essentially leaving my core contention unresolved).  The authors provide some reasonable defense of (although again, not fully resolving) my more minor comments.

Prior to their response I was _absolutely indifferent_ to this paper (genuinely a 3.5).  Now I am happy with very weakly voting for acceptance (4), but I cannot push my score higher because I just do not think are enough contributions in the paper.  The missing comparisons to other baselines/alternatives (also cf. Reviewer eFzT) does not help because I am left wondering how generalizable the insight that is in the paper is.  To drive this paper to the next level I would love to see some more analysis of _why_ and _how_ spectral coverage is key (mechanistically) and the impact on the learning dynamics.  Good luck.

**Limitations:**

No societal impacts.

No substantial technical limitations not already discussed in weaknesses.

**Paper Formatting Concerns:**

None.

**Quality:**

3

**Strengths And Weaknesses:**

# Summary
I think this is fundamentally a solid paper. The contribution is small, but neat and fairly well-motivated. I personally always found the initializion of the timescales and eigenvalues irritating, and so this is a nice exploration of that. The paper itself is well prepared, reads easily with nice figures, and has few typographical errors. I do think the contribution is on the smaller side however. Not insufficient for publication, but nothing to rave over. I err towards acceptance, and am very open to increasing my score should the authors convince me the contribution is larger than I initially give it credit (and answer the few questions I have).
# Strengths
1. The initialization seems to improve performance on real world and simulated tasks.
2. The little theory-esque blocks are nice, helping to concretely isolate and define anything that is mathematically non-trivial, requires precision, or that is difficult to explain in text.
3. Some meaningful ablations and further probes were performed which do flesh out the methods reliability and interest.
4. Something that is not commented on is that S4D-DFouT is a _drop-in replacement and simplification_ into existing codebases and should yield performance improvements across the board.
5. Performance seems less sensitive to $\eta$ than it is to $\Delta$ (making hyperparameter tuning easier).
6. I like tying the initialization across layers (cf. Equ. (11)).

# Weaknesses
1. _I stress to the authors I am very open to changing my mind on this point._ A tendency with initialization-focused papers I don’t like is that: you open with saying you’ll advance the understanding of the frequency-domain computation of these models, propose a new initialization, and then quote improved performance metrics on tasks (that you identify yourself as being flawed in key ways) as back-justifying whatever tweak to an initialization scheme you make.  There is a “gap” between the initialization and the final results that is nearly always uncommented on, training, and specifically, training dynamics. It always just leaves me wanting more, especially when presented with snippets like Figure 1 that begin to unpick the explainability and mechanistic interpretation of these models. I would love to see some slightly more incisive experiments (or justification from the authors that this exists)
    - Mechanistically how the initialization leads to better models through training?
    - Can we design a task with known frequencies and show that poles approach known poles more quickly and reliably across pole distributions?
    - Can we look at raster-scan like tasks (sCIFAR) and preferentially add poles around the image width multiples?
    - Is there a way we can use the frequencies present in the image to better initialize models or compare learned frequencies to (beyond simply the image width)?
    - Inspired by the better distribution of frequencies, is there a way to design even better models for multiple sampling frequencies given the Nyquist/Hermitian observations (cf. SC35)?
    - What is the distribution of learned eigenvalues across layers?  As the computation is fundamentally distributed across layers?
    - If we were to measure the total change of eigenvalues across training (or limit it with a specific trust region) do you get better performance?  Ie., is it the location of the poles themselves, or something rooted deeper in the dynamics of learning multiple eigenvalues and actually it is just the _spacing_ between eigenvalues (as opposed to their value) that makes the learning dynamics more effective?

    These are the sort of questions that I think fill in some of the gap between initialization and raw results.

    I will highlight you go further than a lot of papers (Figure 1, Figure 5 etc.), but I confess I am still left wanting more. To me, this paper feels like: “here is a way you can initialize that seems to lead to better performance” with some motivational text interspersed throughout. This deficiency makes me slightly less enthusiastic about the “completeness” or “togetherness” of the contribution.  I invite the authors to comment on this because I am very open to removing this from my evaluation.

2. Why is S5 excluded as a diagonal variant in Table 1?  This is a fairly major point because S5 does use a diagonal parameterization, and outperforms DFouT-initialized models in most cases.

# Minor Weaknesses.
1. Some typos/errata:
    - Line 216: absis.
    - Throughout (e.g., Line 337): Some of the references to the appendix are incorrect.
    - Throughout (e.g., Table 2): Please try to avoid in-lined floats, they are very bad for the readability of the document.
2. There are a few unsubstantiated claims throughout the paper:
    - Line 175: What does this have to do with “tuning” and/or “interoperability”; what does interoperability even refer to?
    - Line 215: why is it uniquely a universal function approximator where other initializations aren’t?
3. I don’t love punting the one result you apparently do worse on to the supplement, I would like to see results and discussion moved to the main. Especially when psCIFAR probably has the most interesting frequency requirements of the experiments.  It would be a very interesting experiment to use psCIFAR to explore the frequency concentration, because you could use a regular permutation (or even the existing permutation) to explore if the frequencies required for reconstruction are tied to the permutation (because the idea of “locality” has been permuted).
4. I don’t actually understand why you claim it harms performance, it seems like DFouT does the best in Table 3?  Unless there is something I’m misunderstanding?
5. I don’t understand the middle panel of Figure 4.  What is the distortion?  I think it has a key meaning that I simply cannot grok from the text.  It seems like smaller timescales with Lin (the default initialization) do better?  What happens if $\Delta = 10e-4$?

---

> ### Author Rebuttal · Authors · 2025-07-31
>
> Dear reviewer, thank for your time in assessing our paper, and for the valuable comments.
>
> > **Reviewer[YT22] Point #1** : "Mechanistically how the initialization leads to better models through training?"
>
> We understand the concerns of a circular justification—“better initialization implies better performance, therefore the initialization must be inherently superior.” Here are two findings we observed:
>
> **- Preservation of the initial spectral distribution.** We depict in Figure S9 in the Supplementary Material, the final distribution of the eigenvalues of A layer-by-layer after training in sCIFAR. We observe that in all initializations, the angles of the poles (resonant frequencies) remain largely unchanged from their initial values. However, poles exhibit a significant change in the decay, presenting a smaller radius after training. At the end of the training, S4D-Lin still shows the entanglement between decay and frequency,  resulting in spiral-like clustering of the poles; S4D-Inv spreads poles evenly around the circle; and S4D-DFouT keeps the original half-plane distribution of poles throughout training.  This observation indicates that the model preserves its initial spectral basis and primarily refines modal amplitudes during optimization. Consequently, we argue that a comprehensive frequency coverage at initialization contributes ultimately to elevated final performance. This reasoning thereby avoids the circular argument.
>
> **- Enhanced utilization of the spectral domain.** We also observe in Figure S9 that compared to S4D-Lin and S4D-Inv, the poles in DFouT that contribute most significantly (as measured by their $\mathcal{H}_\infty$-norm) are distributed more uniformly across both quadrants of the upper half-plane, rather than being concentrated at low frequencies. Consequently, DFouT more effectively leverages high-frequency modes that capture fine-grained input features.
>
> Although these observations do not fully explain the learning dynamics, they contribute to a more precise understanding of the role of initialization beyond mere final performance. We appreciate the reviewer’s suggestion to undertake this deeper analysis.
>
> > **Reviewer[YT22] Point #2** : "Can we design a task with known frequencies and show that poles approach known poles more quickly and reliably across pole distributions?"
>
> Actually, the Continuous Copying task was motivated by a similar reason. In the Continuous Copying, we know a priori the frequencies (fundamental and harmonics) required to generate a _delay_ kernel exhibiting a spike at the desired $\tau$.
>
> One insight we have observed during training—as introduced in Point 1—is that the pole angles (resonant frequencies) remain largely unchanged throughout training. In this context, we also note that succeeding in the Continuous Copying task requires spectral coverage of those fundamental and harmonic frequencies from initialization. However, as illustrated in Figure 3, the spectral distribution of previous S4D initializations is strongly influenced by $\Delta$, making it difficult for this initialization to learn that kernel unless the order $N$ of the model is sufficiently large (we tested up to $N=1024$). In the case of S4D‑Lin, the hyperparameter $\Delta$ dictates the pole distribution: small $\Delta$ values cluster poles at low frequencies, while large $\Delta$ values suffer from aliasing. By contrast, S4D‑DFouT is more robust because it provides a uniform distribution of poles across the spectrum, providing initial coverage of these relevant frequencies otherwise hard to optimize during training.
>
> > **Reviewer[YT22] Point #3-#4** : "Can we look at raster-scan like tasks (sCIFAR) and preferentially add poles around the image width multiples?"
>
> Thank you for the insightful question. While our primary aim was to design a data‑agnostic initialization that avoids some of the pitfalls of prior schemes, the question of using the model insights to exploit the spectral structure of the data is very on the point.
>
> To test this hypothesis, we have designed an initialization **S4D-Spectrum** where the real part is preserced from DFouT and the imaginary part matches the discretize spectrum distribution (normalized) of the data.
>
> | Method                 | sCIFAR           | SC35 16kHz         | SC35 8kHz          |
> |------------------------|------------------|--------------------|--------------------|
> | S4D-Spectrum           |      67.17            |        96.10            |          91.31          |
>
> **Results.**  Regarding sCIFAR, its spectrum presents a fundamental frequency $\Omega=2\pi/32$ and clear harmonics at multiples of that fundamental (Figure S7 in the Supplementary Material). However, using this distribution S4D-Spectrum results in a noticeably less performant model. The insights we provided in Section 5.2 of the main paper can be useful in explaining such behaviour. In that experiment, we measure how much each SSM actually contributes to the output by computing its H-norm (Figure 5, right). We observe that the SSMs operating in the first layer close to the fundamental frequency, present noticeably larger H-norms. However, in successive layers, the frequencies of importance move toward lower bands.  The signal experiences progressive temporal mixing throughout the layers toward lower frequencies. In this regard, placing poles only at the marked frequencies in -all layers- does not benefit to the model. It seems to us, though, that setting a temperature hyperparameter to smooth that distribution could lead to better results.
>
> > **Reviewer[YT22] Point #5** :  "Inspired by the better distribution of frequencies, is there a way to design even better models for multiple sampling frequencies given the Nyquist/Hermitian observations (cf. SC35)?"
>
> We extended the experiment by training S4D-Spectrum on SC35. In this setting, the spectrum of the data is more evenly distributed, yielding a more uniform pole coverage compared to sCIFAR. As a result, we observe performance on par with other initialization methods. S4D-Spectrum already incorporates Nyquist and Hermitian symmetry. We design its frequencies to span only the positive half of the spectrum—sampling the DFT from \([0,\pi]\)—therefore, it does not suffer from aliasing.
>
> > **Reviewer[YT22] Point #6** :  "What is the distribution of learned eigenvalues across layers? As the computation is fundamentally distributed across layers."
>
> As previously introduced in Point \#1, we already depict in Figure S9 in the Supplementary Material the final distribution of the eigenvalues of A layer-by-layer after training in sCIFAR for the different initialization schemes. Our major observation is that in all initializations, the angles of the poles (resonant frequencies) remain largely unchanged from their initial values. However, poles exhibit a significant change in the decay, presenting a smaller radius after training. In the particular task of sCIFAR, we believe that some poles are learned to decay rapidly to fine-control the initial coefficients of the kernel, which, as reported in Figure 5 (left), are particularly relevant for performance in this task (especially the first 32 coefficients).
>
> > **Reviewer[YT22] Point #7** : "If we were to measure the total change of eigenvalues across training (or limit it with a specific trust region) do you get better performance?
>
> Empirically —as commented in Point \#1—we find that the vast majority of change on the poles occurs in the decay (radius); and even due to the time limitation, we could not complete the experiment setting a trust region, we expect no substantial improvement in the accuracy. Our observations suggest that ensuring good spectral coverage upfront is key.
>
> > **Reviewer[YT22] Point #8** : "Why is S5 excluded as a diagonal variant in Table 1? This is a fairly major point because S5 does use a diagonal parameterization, and outperforms DFouT-initialized models in most cases."
>
> We agree that the composition of Table 1 is not fair to S5 [6], since this method also uses a diagonal state matrix. We will complete Table 1 with S5 results, but also include our proposed initialization with S5 architecture. However, we would like to point an important observation concerning this method that highlights our main findings. Namely, the authors of S5 also reported in Table 7 of their work, results for different initializations as S5-Lin and S5-Inv in addition to S5 (which follows HiPPO-N matrix)—results that we have also collected in our Table S7. Interestingly, S5-Lin again underperforms on tasks as PathX-128, as the authors noted in their Appendix E.3. Therefore, rather than focusing on overall performance, we propose rewriting that table to highlight that previous Fourier-based initializations such as S4-Lin or S5-Lin have repeatedly underperformed in the literature due to some of the limitations we identify in this paper.
>
> > **Reviewer[YT22] Question #1** :   "Doesn’t the locality of the receptive field influence the decay rate?
>
>  We agree that the discussion in the decay is also very relevant for the work. We compiled some insights about the decay in the Supplementary Material (Figure S9, commented in Point \#6); yet those conclusions should have a space in the main document.
>
> Yes, if the eigenvalues responsible for capturing 32-pixel do not decay (i.e., if they preserve energy), then we would also see repeating peaks in the kernel—not only at 32, but at 64, 96... Indeed, this is exactly what we observe in the kernels of Figure S7. However, those later peaks are much attenuated due to the decay of the corresponding poles. We also provide an explanation for that behaviour. Learning a locally constrained receptive field can be beneficial for these models, as it mimics how CNNs operate: where early convolutional layers specialize in detecting localized features, while deeper layers progressively integrate these patterns into more abstract global representations.

---

> > ### Comment · Reviewer_YT22 · 2025-08-05
> >
> > Thank you to the authors for their comment.  Your comments have helped some of my concerns.  I'm now happy giving a 4 (I was borderline between 3 and 4 prior to rebuttal).  I still stand by my overall thrust that this work observes that initializing close to the final solution improves performance and does offer a neat way of initializing there, but that it lacks more incisive analysis on why this is useful or how to design better models etc.  For this reason, I strongly consider myself "borderline accept".  I would be okay if this paper were included, but I do not think there is quite enough there to be enthusiastic about acceptance.  To improve this paper, I would love to see more developed: examination of the learning dynamics, elements of mechanistic interpretability on why magnitude/phase are important for different features, designing better architectures/layers with this knowledge in mind etc.  Good luck.

---

### Official Review · Reviewer_V8Gu · 2025-07-02

**Clarity:** 3
**Significance:** 3
**Originality:** 4
**Rating:** 5
**Confidence:** 3

**Summary:**

This paper investigates diagonal variants of State Space Models (SSMs), focusing on how initialization affects their ability to capture long-range dependencies. The authors analyze these models through a frequency-domain lens and highlight that existing initialization schemes suffer from spectral aliasing and sensitivity to discretization. To address this, they propose S4D-DFouT, a novel initialization strategy that places poles directly in the discrete Fourier domain, ensuring uniform spectral coverage and robustness. Empirical results on the Long Range Arena benchmark show that this method improves performance and enables training from scratch on previously unsolved tasks like PathX-256.

**Questions:**

- Can the proposed initialization strategy (S4D-DFouT) be extended to low-rank or more general structured SSMs beyond the diagonal case? If so, what challenges or adjustments would be required?

- What types of tasks or data characteristics might cause S4D-DFouT to underperform? Are there known limitations where this initialization is less effective?

- Could you provide a comparison of computational cost (e.g., training time, memory usage) versus accuracy to better illustrate the efficiency gains of S4D-DFouT relative to other models?

**Ethical Concerns:**

["NO or VERY MINOR ethics concerns only"]

**Final Justification:**

I believe the rebuttal was convincing and I look forward to  seeing all the revisions mentioned by the author for the final paper.

**Limitations:**

yes

**Paper Formatting Concerns:**

no issue found.

**Quality:**

3

**Strengths And Weaknesses:**

Strengths

- Clear motivation and direction: The paper’s focus on model initialization as a path to improving diagonal SSMs is timely and relevant. The emphasis on long-range dependency modeling aligns well with current research trends and opens avenues for future work.

- Well-written and theoretically grounded: The paper is clearly written and includes a solid theoretical foundation, particularly in Section 3, where the authors provide a detailed frequency-domain analysis of diagonal SSMs.

- Novel method (S4D-DFouT): The proposed initialization scheme, S4D-DFouT, is theoretically justified and introduces a principled way to achieve uniform spectral coverage. It is a simple yet impactful contribution.

- Strong empirical results: The method outperforms existing diagonal SSM variants across multiple tasks in the Long Range Arena benchmark (Table 1) and is the first to train PathX-256 from scratch without pretraining.


Weaknesses:

- Limited ablation beyond SSMs: While the paper provides thorough comparisons within the diagonal SSM family, it lacks ablations against non-SSM models like Transformers. Such comparisons would help clarify the broader applicability of the proposed method.

- Motivation could better connect to practical use cases: Although the motivation is theoretically compelling, it would benefit from examples or discussion showing potential industrial applications or real-world scenarios where spectral control via initialization is especially valuable especially compared to other models including transformers, MoE.

- Lack of compute resource reporting: The paper does not report training time, memory consumption, or hardware setup. These details are critical for evaluating claims of efficiency and scalability. I could not find these details mentioned in the paper.

- Environmental impact unaddressed: The authors could strengthen the paper by reporting on the carbon footprint or potential energy savings of their method compared to more compute-intensive baselines like Transformers.

---

> ### Author Rebuttal · Authors · 2025-07-29
>
> Dear reviewer thank for your time in assessing our paper, and for the valuable comments.
> > **Reviewer[V8Gu] Point #1** : Limited ablation beyond SSMs: While the paper provides thorough comparisons within the diagonal SSM family, it lacks ablations against non-SSM models like Transformers. Such comparisons would help clarify the broader applicability of the proposed method.
>
> In fact, in Table S4 in Supplementary Material, we already include a comparison on Long Range Arena tasks with standard Transformer architecture and some of its variants as Performer, Reformer, and MEGA. In the revised version of the paper, we plan to include these variants in the main article, Table 1.
>
> > **Reviewer[V8Gu] Point #2** : Motivation could better connect to practical use cases: Although the motivation is theoretically compelling, it would benefit from examples or discussion showing potential industrial applications or real-world scenarios where spectral control via initialization is especially valuable, especially compared to other models, including transformers, MoE.
>
> We actually provide two tasks that pertain to industrial applications or real-world scenarios. First, we evaluate on the Speech Commands dataset (SC35) for audio signal classification, where our proposed initialization scheme achieves state-of-the-art in zero-shot resampling (Table 2). Second, we assess physiological signal prediction using the BIDMC dataset, targeting respiratory rate, heart rate, and blood oxygen continuous time-series data. As shown in Appendix Table S3, a variant of our decoupling method also achieves state-of-the-art results in this domain.
>
> Prior work [1] notes that SSMs often underperform in tokenized or language tasks but excel in continuous domains such as signals or time series. Therefore, we are confident that our method has a big potential in industrial applications.
>
> [1] - Dao, Tri, and Albert Gu. "Transformers are SSMs: Generalized Models and Efficient Algorithms Through Structured State Space Duality." International Conference on Machine Learning. PMLR, 2024.
>
> > **Reviewer[V8Gu] Question #1** :Extension to Low-rank: Can the proposed initialization strategy (S4D-DFouT) be extended to low-rank or more general structured SSMs beyond the diagonal case? If so, what challenges or adjustments would be required?
>
>  Introducing off‑diagonal or low‑rank components does make the spectrum, and thus mode behavior, less interpretable than in the purely diagonal case. In structured/low‑rank SSMs as S4, modes become linear combinations of eigenvectors, and low‑rank terms $UV^T$ perturb the poles. As a result, it is difficult to attribute specific frequency behavior to individual parameters—for example, pinpointing which row captures a given frequency or understanding how the amplitude and phase of that mode are governed by $C_n\bar{B}_n$. So, to answer the question, we believe the extension is possible, but we do not know how to do it in an "explainable" way.
>
> > **Reviewer[V8Gu] Question #2** :Discussion on underperformance: What types of tasks or data characteristics might cause S4D-DFouT to underperform? Are there known limitations where this initialization is less effective?
>
> In Section 6, we briefly outline the current limitations of SSMs—a discussion we will expand in the final draft. We show that many Long Range Arena benchmarks regarding sequential images can be “solved” by exploiting local spectral biases (for example, capturing frequencies corresponding to pixel offsets within image rows), rather than genuine long‑range modeling. In contrast, tasks that span the full frequency spectrum—such as ListOps or permuted sequential images like psCIFAR (see the flattened spectrum in Supplementary Figure S7)—remain far from solved. Although SSMs still outperform other architectures as Transformers on the latter problems, these are far from being considered solved.
>
> > **Reviewer[V8Gu] Question #3**:Computational cost: Could you provide a comparison of computational cost (e.g., training time, memory usage) versus accuracy to better illustrate the efficiency gains of S4D-DFouT relative to other models?"
>
> A brief mention of the computational cost is done in the introduction, but we would increase the information in the manuscript. Diagonal SSMs present a significant cost reduction w.r.t. non-diagonal variants. As deeply inspected in [2], by materializing the Vandermonde matrix of A, the computation cost of computing the kernel in diagonal SSMs can be obtained in O((N+L) log2(N+L)) arithmetic operations. Our work builds upon these efficient diagonal variants; we do not propose architectural modifications, but rather aim to provide a mechanistic interpretation of those models. As such, the computational/memory cost remains unchanged.
>
> [2] Gu, Albert, et al. "On the parameterization and initialization of diagonal state space models." Advances in Neural Information Processing Systems 35 (2022): 35971-35983.
>
> We appreciate the reviewer’s comments, which helped us clarify both the motivation and broader implications of our work. We believe these revisions better position the paper’s contributions and respectfully invite reevaluation based on the new context.

---

> ### Comment · Reviewer_V8Gu · 2025-08-02
>
> Thanks to the explanation provided, I believe making the motivation more clear in the introduction by including industrial examples that can better benefit from your model can attract broader audience.
> Also, I believe the comparison with other models including transformers should not be in supplementary and it should appear in the main paper.
> Overall, I find the paper interesting and think it has its own merits.
> I have modified my score

---

### Note · Authors · 2025-08-12

Dear AC,

We would like to thank the reviewers for their dedication and the positive outcome during the rebuttal. The changes introduced have certainly contributed to improving the manuscript.

As a final comment, we would like to draw your attention to the reviewer’s eFzT comments, who mentions that the contributions of this work are close to the border of acceptance, while leaving the final decision to the AC's criteria. During the rebuttal, we highlighted that the work we present here is not just an Initialization scheme for SSMs, but as we intended to reflect from the title and along the sections of the paper and the experimentation—especially in the ablations presented in Sections 5.1 (importance of pole distribution) and 5.2 (learning biases on sequential images)—  the main narrative of this work is 'Uncovering the mechanistic learning behaviors of SSMs usign a fequency analaysis'. We show the learning biases these models exploit to learn in tasks considered complex as Long Range Arena (LRA). Furthermore, we validate these claims, showing that they can be exploited to learn from scratch on PathX-256, a task considered too hard to learn without pre-training in the literature. The reviewer also points out that LRU initialization achieves better performance in 50% of LRA tasks, but we remarked in the rebuttal that LRU settings for each task are two or three times bigger at least in the number of poles in the mentioned tasks.

As a summary, rather than showing sota results in all tasks, this work aims to provide insights on how the learning occurs on time-invariant diagonal SSMs (analysis of the lobes, spectrum coverage, impact of the kernel length, etc). This is reflected along the paper until Supplementary Figure S9, the Figure that the reviewer acknowledges and points out to be:

> This type of analysis indeed contributes meaningfully to the field. However, if the authors feel that this analytical perspective is their primary contribution, then the paper should be positioned as such.

We stress that Figure S9 was placed in the Supplementary Material only due to space constraints, and a simplified version of it was presented in Figure 3. Its insights are supported throughout the main paper, ending in the design rationale behind the proposed initialization. We believe this analysis meaningfully advances the understanding of SSM behavior.

---

### Decision · Program_Chairs · 2025-09-17

**Decision:**

Accept (poster)

**Comment:**

This paper proposes a novel initialization scheme for diagonal state space models. The reviewers generally found the paper to be well-written, and the contribution appears to be novel, conceptually motivated, and the method empirically outperforms several baselines.

Hopefully the authors can address in their final version some of the concerns about significance. Reviewers were particularly interested in understanding the mechanics of the learning process, both to explain the performance benefits of the proposed method and to potentially extend the impact of the frequency-based analysis to other aspects of the model (like features and/or architectures). It would be great to see some additional baselines in the experiments as well, as suggested by several reviewers.